# Oxytocin signaling in the medial amygdala is required for sex discrimination of social cues

**Shenqin Yao[†], Joseph Bergan[‡], Anne Lanjuin[§], Catherine Dulac***

Department of Molecular and Cellular Biology, Center for Brain Science, Howard Hughes Medical Institute, Harvard University, Cambridge, United States

**Abstract** The neural control of social behaviors in rodents requires the encoding of pheromonal cues by the vomeronasal system. Here we show that the typical preference of male mice for females is eliminated in mutants lacking oxytocin, a neuropeptide modulating social behaviors in many species. Ablation of the oxytocin receptor in aromatase-expressing neurons of the medial amygdala (MeA) fully recapitulates the elimination of female preference in males. Further, single-unit recording in the MeA uncovered significant changes in the sensory representation of conspecific cues in the absence of oxytocin signaling. Finally, acute manipulation of oxytocin signaling in adults is sufficient to alter social interaction preferences in males as well as responses of MeA neurons to chemosensory cues. These results uncover the critical role of oxytocin signaling in a molecularly defined neuronal population in order to modulate the behavioral and physiological responses of male mice to females on a moment-to-moment basis.

DOI: https://doi.org/10.7554/eLife.31373.001

**\*For correspondence:**
dulac@fas.harvard.edu

**Present address:** [†]Allen Institute for Brain Science, Seattle, United States; [‡]Department of Psychological and Brain Sciences, University of Massachusetts, Massachusetts, United States; [§]Harvard TH Chan School of Public Health, Boston, United States

## Introduction

A fundamental goal of neuroscience is to understand how brain circuits control behavior. New advances in genetic, imaging and functional approaches applied to small and large brains are uncovering wiring diagrams and neural activity ensembles underlying specific behaviors at increasingly high resolution. Proper regulation of the functional properties of these neural circuits requires cohorts of still poorly understood neuromodulators, such as biogenic amines and neuropeptides (*Bargmann and Marder, 2013*; *Marder, 2012*). Released from the axons, dendrites, and cell bodies of specific neuronal subpopulations, neuropeptides signal on time scales ranging from seconds to minutes and hours, and act on local microcircuits as well as on diffuse targets distributed throughout the brain. Thus, neuromodulation adds temporal and spatial dynamics to the function of neural circuits according to the animal's external environment and internal physiological state (*Brezina, 2010*; *Hökfelt et al., 2000*).

The vertebrate hypothalamus has emerged as a particularly rich source of neuropeptides. These neuromodulators shape intricate allostatic and behavioral functions such as temperature and energy balance, thirst, hunger, sleep, as well as aggression, reproduction, and parenting. In particular, the nonapeptides oxytocin (OXT) and vasopressin (AVP) represent an important set of neuromodulators that are produced by discrete populations of hypothalamic neurons in response to social signals and according to an animal's physiological state (*Donaldson and Young, 2008*; *Knobloch and Grinevich, 2014*; *Landgraf and Neumann, 2004*; *Veenema and Neumann, 2008*). These nonapeptides are evolutionarily conserved across vertebrate and invertebrate species, and it has been proposed that sex- and species-specific differences OXT and AVP systems may underlie genetic variations in social behavior control (*Bendesky et al., 2017*; *Caldwell, 2017*; *Johnson and Young, 2017*; *Knobloch and Grinevich, 2014*; *Lockard et al., 2017*; *Vaidyanathan and Hammock, 2017*).

**eLife digest** Oxytocin is a hormone that promotes milk production, contractions during childbirth, and many social interactions in humans and other creatures. It has also been implicated in conditions like autism or schizophrenia, which show altered social interactions. Oxytocin is made and released by cells in the brain called neurons. The oxytocin-producing neurons are clustered in a brain region called the hypothalamus, and oxytocin can act over a long distance in the brain or in the body. Many mammals detect chemical signals called pheromones that are involved in social interactions. These chemicals are detected by neurons in a structure within the cartilage of the nose called the vomeronasal organ. Pheromone-sensing neurons in the vomeronasal organ connect with another part of the brain called the medial amygdala. The medial amygdala, in turn, connects with regions of the brain that control behavior.

Mice in particular rely on pheromones for social communication. Male and female mice respond differently to pheromones. Male mice prefer to investigate female mice to other males. The neurons in medial amygdala of male mice also become more active in response to scents from females than from males. Oxytocin is known to act on the medial amygdala, but its exact role in the male's preference for females and their scents is not known.

Now, Yao et al. show that oxytocin controls male preference for interacting with females and their scents by turning on neurons in the medial amygdala. In the experiments, male mice genetically engineered to lack oxytocin do not prefer female mice to other males, and they also appear unable to distinguish male and female scents. These mice also have less activity in the neurons of the medial amygdala when exposed to females and their scents. Directly manipulating these neurons and the oxytocin receptors on them also altered sex-preferences in male mice.

The experiments show that oxytocin alters the behaviors of male mice in response to females or their scents by manipulating a specific set of brain cells. More studies of these cells or their interactions with oxytocin might help scientists understand oxytocin-liked diseases that impair social interactions or develop new treatments for conditions like autism or schizophrenia.

DOI: https://doi.org/10.7554/eLife.31373.002

Genetic models of mice deficient in OXT or in OXT receptor (OXTR) have helped uncover the requirement of OXT signaling in myriad social behaviors including recognition of familiar and novel conspecifics (*Ferguson et al., 2001*, Ferguson et al., 2000*Ferguson et al., 2000*; *Takayanagi et al., 2005*; *Wersinger et al., 2008*), parental care (*Takayanagi et al., 2005*), social interactions of female with male mice during the estrous cycle (*Nakajima et al., 2014*), aggression (*Harmon et al., 2002*), and anxiety-related behaviors in male mice (*Li et al., 2016*). OXT can act directly via OXTR signaling on postsynaptic neurons to alter the activity of key components of neural circuits regulating social behaviors as seen in circuits controlling female social approach to males at specific phases of the estrus cycle (*Nakajima et al., 2014*), anxiety-related behaviors in male mice (*Li et al., 2016*), and maternal responses to pup calls (*Marlin et al., 2015*). OXT has also been shown to modulate social behaviors indirectly by promoting the processing of social odor cues in the main olfactory bulb via top-down projections from the anterior olfactory nucleus (*Oettl et al., 2016*), by acting on other neuromodulatory systems, for example regulating the activities of dopamine neurons (*Xiao et al., 2017*), facilitating norepinephrine release in the olfactory bulb to promote social recognition (*Dluzen et al., 2000*; *Dluzen et al., 1998*) or reinforcing social interaction with conspecifics by regulating presynaptic serotonin release in the nucleus accumbens (*Dölen et al., 2013*).

In mice, genetic and surgical ablation of the vomeronasal organ (VNO), a specialized organ required for the detection of pheromones, results in atypical behaviors towards conspecifics, with female and male mutants displaying sexual behaviors indiscriminately towards conspecifics of both sexes, suggesting that the vomeronasal system plays a critical role in decoding sex-specific chemosensory cues (*Kimchi et al., 2007*; *Liman et al., 1999*; *Stowers et al., 2002*). The medial amygdala (MeA), located two-synapses downstream of VNO chemosensory neurons, and projecting to hypothalamic areas controlling innate behavioral responses, is an important brain region implicated in the processing of social cues (*Kevetter and Winans, 1981a*; *Kevetter and Winans, 1981b*; *Petrovich et al., 2001*). The MeA displays specific, topographically organized, and sexually

dimorphic responses to chemosensory cues (*Bergan et al., 2014*; *Choi et al., 2005*; *Lehman et al., 1980*). Functional manipulation of the MeA has been shown to affect various social behaviors, such as aggression, mating, social-grooming, and social recognition (*Choleris et al., 2007*; *Ferguson et al., 2001*). OXT signaling in the MeA is critical for mice to distinguish familiar from novel conspecifics; however, the involvement of OXT in modulating other vomeronasal functions, as well as the mechanisms by which OXT sculpts the discrimination of socially relevant cues are not yet fully understood (*Choleris et al., 2007*; *Ferguson et al., 2001*; *Gur et al., 2014*).

Here, we investigated OXT action on neural circuits regulating sex-specific behavioral responses to conspecific signals. We found that, during social interactions, OXT mutant males do not display the preference for female conspecifics that is typical of wild-type males, and are impaired in chemosensory discrimination of sex-specific cues. To dissect the underlying cellular components of OXT function in behavioral sex-discrimination, we conducted genetic ablation as well as virus-mediated manipulation of OXTR, and demonstrate the critical role of OXT signaling in a sub-population of MeA neurons expressing the steroid converting enzyme aromatase. We further show that impaired chemosensory discrimination in OXT mutants correlates with altered chemosensory response profiles in the MeA. Moreover, acute modulation of OXT signaling in adults is sufficient to alter sensory representation in the MeA and sex discrimination in social interactions. Our study thus reveals that acute neuromodulatory function of OXT in a single, molecularly defined, population of neurons shapes the sensory representation of the MeA and is critical for tuning an animal's preference for male versus female social cues.

## Results

### OXT signaling is required for the discrimination of sex-specific cues in males but not in females

Social behavior responses are fine-tuned according to the species, gender, and endocrine status of individual animals. To investigate a potential role of OXT in the modulation of sex-specific social preference behaviors, we compared the preference of C57BL/6J OXT knockout male mice ($Oxt^{-/-}$) and wild-type littermate controls ($Oxt^{+/+}$) in investigating female versus male conspecifics (*Figure 1A*). Subject male mice were group-housed with littermates of the same sex after weaning, individually housed for one week before behavioral tests, and sexually naïve at the onset of testing. After an initial habituation period of 10 min in a 3-chamber paradigm with an empty wire cup in each side chamber (*Yang et al., 2011*), $Oxt^{+/+}$ and $Oxt^{-/-}$ mice were allowed to explore the arena with a novel male confined in the wire cup in one side chamber and a novel female confined in the wire cup in the other side chamber. The female and male sides were randomized between experiments. Both stimulus mice were confined in wire cups that permit the exchange of visual, chemosensory, and acoustic signals between subject and stimulus mice.

The amount of time subjects spent adjacent to each enclosure cage was quantified, and a 'social preference score' (see Materials and methods) was calculated that ranges from −1.0 (all time spent next to the male cage) to 1.0 (all time spent next to the female cage). Data show that control $Oxt^{+/+}$ male mice spent significantly more time investigating females than males (female interaction zone = 192.8 ± 12.2 s; male interaction zone = 128.0 ± 10.8 s; p<0.001, t test; n = 10) resulting in a positive social preference score that is significantly higher than one obtained with empty wire cups (0.21 ± 0.03 vs. −0.001 ± 0.05; p<0.001, t test; n = 10; *Figure 1B*). By contrast, $Oxt^{-/-}$ male mice displayed little preference for investigating female over male conspecifics (male interaction zone = 105.1 ± 13.2 s, female interaction zone = 121.8 ± 13.8 s; p=0.39, t test; n = 10) and had a social preference score that was not significantly different from one obtained with empty wire cups (0.08 ± 0.05 vs. 0.03 ± 0.05; p=0.52, t test; n = 10; *Figure 1B*). When tested with empty enclosure cages, both $Oxt^{+/+}$ and $Oxt^{-/-}$ males spent equal time in the two interaction zones ($Oxt^{+/+}$: empty male interaction zone = 141.7 ± 13.9 s, empty female interaction zone = 137.8 ± 7.7 s; p=0.81, t test; $Oxt^{-/-}$: empty male interaction zone = 108.8 ± 7.7 s, empty female interaction zone = 115.9 ± 7.5 s; p=0.52, t test). Thus, our results suggest that loss of OXT alters the normal preference of male mice to interact with female rather than male conspecifics.

Next, we measured the preference of OXT mutant mice for investigating female versus male bedding, a rich source of conspecific chemostimuli. In this preference paradigm (*Figure 1C*), individually

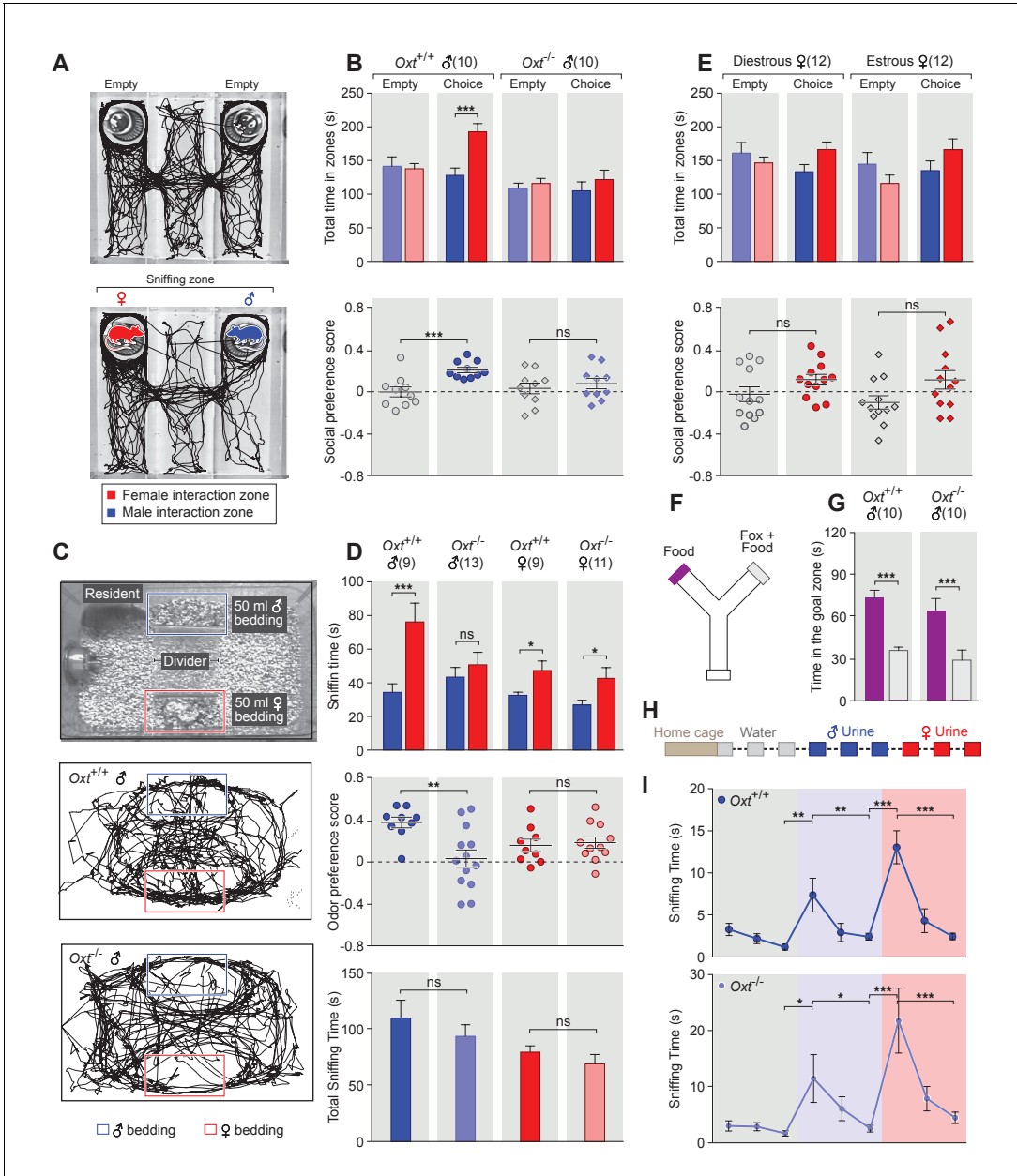

**Figure 1.** OXT signaling is required for male mice to discriminate female and male conspecifics. (A) A 3-chamber-social-investigation paradigm testing social preference. Representative traces of a male subject mouse habituating in the 3-chamber paradigm with empty enclosure cups (Upper), and then exploring male and female mice confined in the enclosure cups (Bottom). (B) Comparison of time spent by $Oxt^{+/+}$ and $Oxt^{-/-}$ male mice in each interaction zone (Upper, t test) and social preference score (Bottom, t test). ***p<0.001, ns, not significant. (C) A chemostimuli preference paradigm and representative traces of $Oxt^{+/+}$ and $Oxt^{-/-}$ male mice. (D) Comparison of time spent by $Oxt^{+/+}$ and $Oxt^{-/-}$ male and female mice investigating each odor cue (Upper, t test), odor preference score (Middle, t test) and total time spent investigating odor cues (Bottom, t test). ***p<0.001, **p<0.01, *p<0.05, ns, not significant. (E) Comparison of time spent by diestrous and estrous female mice in each interaction zone (Upper, t test) and social investigation preference score (Bottom, t test). (F–G) Y-maze paradigm testing the willingness of mice to investigate goal zones with food vs. fox urine spiked food. t test, ***p<0.001. (H) Odor habituation by repeat presentation of one urinary odor and dishabituation by a second urinary odor. (I) Time spent by mice investigating each odor indicates that both $Oxt^{+/+}$ and $Oxt^{-/-}$ male can be habituated by the urine from one sex and then dishabituated by the urine from another sex. Fisher's LSD test, ***p<0.001, **p<0.01, *p<0.05.

DOI: https://doi.org/10.7554/eLife.31373.003

The following figure supplement is available for figure 1:

**Figure supplement 1.** Conspecific odor cue preference of female mice in different estrous stages.

DOI: https://doi.org/10.7554/eLife.31373.004

caged naïve $Oxt^{+/+}$ and $Oxt^{-/-}$ male littermates were presented in their home cages with male and female bedding separated by a divider. Subject mice were allowed to investigate the two types of bedding for a total of 5 min. As expected, $Oxt^{+/+}$ males spent a significantly longer time on average investigating female than male bedding (male bedding = 34.3 ± 5.1 s, female bedding = 76.3 ± 11.2 s; p<0.001, t test; n = 9), whereas $Oxt^{-/-}$ male mice presented lost preference in investigating female chemostimuli (male bedding = 43.5 ± 5.7 s, female bedding = 50.8 ± 7.4 s; p=0.46, t test; n = 13; *Figure 1D*). Accordingly, the mean odor preference score (see Materials and methods) of $Oxt^{+/+}$ male mice (0.38 ± 0.05) was significantly higher than that of $Oxt^{-/-}$ male mice (0.03 ± 0.08; p<0.01, t test; *Figure 1D*). However, the overall time spent investigating either bedding was similar between $Oxt^{+/+}$ and $Oxt^{-/-}$ male mice (110.6 ± 15.8 s vs. 94.24 ± 10.3 s; p=0.37, t test; *Figure 1D*), suggesting that the loss of male preference for female cues is not simply due to general loss of interest in socially relevant chemostimuli. Our results thus demonstrate that OXT signaling is required for male mice to display normal preference for female mouse chemosensory cues.

Next, we investigated gonadally intact females for the effect of OXT signaling on preference for conspecific cues. Socially naïve C57BL/6J $Oxt^{+/+}$ and $Oxt^{-/-}$ female mice were group-housed since weaning with littermates of the same sex and genotype, and individually caged for one week before being tested for their preference in investigating female versus male bedding. In contrast to the observations made with male mice, loss of OXT did not cause significant changes in odor preference in females. Both $Oxt^{+/+}$ and $Oxt^{-/-}$ female littermates spent slightly, but statistically significant, longer durations investigating female bedding than male bedding ($Oxt^{+/+}$: male bedding = 32.6 ± 1.8 s, female bedding = 47.4 ± 5.7 s, n = 9; $Oxt^{-/-}$: male bedding = 26.9 ± 2.8 s, female bedding = 42.7 ± 6.4 s, n = 11; p<0.05, t test; *Figure 1D*), and there was no significant difference between the two groups in odor preference score ($Oxt^{+/+}$:0.16 ± 0.06, n = 9; $Oxt^{-/-}$:0.19 ± 0.05, n = 11; p=0.75, t test; *Figure 1D*). Importantly we investigated sex preference at different stages of estrous cycle in wild-type females (*Figure 1—figure supplement 1*) in the 3-chamber social interaction assay that allowed visual, auditory and chemosensory interactions (*Figure 1E*). No significant preference was observed in either estrous or di-estrous C57BL/6J female mice (t test for comparisons of time spent in interaction zones and beddings, and for comparison of preference scores and total sniffing time). Based on these data, we focused our subsequent investigation on the robust behavioral preference observed by males for female cues.

To explore a more general role of OXT in the recognition of animal cues, including heterospecific signals, we utilized a Y-maze paradigm to test the responses of $Oxt^{+/+}$ and $Oxt^{-/-}$ mice to predator cues (*Figure 1F*). After initial habituation in the Y-maze, subject mice were allowed to explore the side chambers (goal zones) of the Y-maze containing only a food pellet on one side and both fox urine and a food pellet on the other side. We measured the time mice spent in each goal zone and found that $Oxt^{+/+}$ (fox +food = 35.8±2.4 s, food = 73.4 ± 5.5 s; p<0.001, t test; n = 10) as well as $Oxt^{-/-}$ (fox +food = 28.9±7.3 s, food = 63.9 ± 9.0 s; p<0.001, t test; n = 10) mice spent significantly less time in the goal zone containing fox urine in additional to a food pellet (*Figure 1G*). Therefore, mice with impaired OXT signaling readily recognize and adapt behaviors in response to predator cues, supporting a specific role for OXT in modulating social interaction of male mice with conspecifics.

The loss of behavioral preference for female cues observed in $Oxt^{-/-}$ males could be due to circuit changes at any stage along the sensorimotor transformation. Consistent with previous findings (*Ferguson et al., 2000*), our results show no significant difference in the combined time that $Oxt^{-/-}$ and $Oxt^{+/+}$ mice spent investigating male and female beddings, suggesting that mice with impaired OXT signaling have similar ability and motivation to investigate conspecific chemical cues. To exclude the possibility that diminished cue preference results from changes in signal detection, we carried out an olfactory habituation and dishabituation assay (*Crawley et al., 2007*; *Yang and Crawley, 2009*) (*Figure 1H*). Individually caged male mice received three successive 2 min presentations of mouse urine samples from one sex at an interval of 1 min, followed with samples from the other sex in three subsequent presentations. Data show that both wild-type and OXT mutant males habituate to repeated presentation of the urine from the same sex, as indicated by reduced duration spent investigating the odor source, and that they subsequently dishabituate to urine from the other sex (Fisher's LSD test, ***p<0.001, **p<0.01, *p<0.05; *Figure 1I*). Thus, mice lacking OXT still have the ability to detect and remember conspecific cues, suggesting that the lack of behavioral bias towards

females in $OXT^{-/-}$ male mice is likely due to abnormal processing of conspecific cues rather than a mere defect in signal detection.

## OXT function in chemosensory discrimination is mediated by the OXT receptor

OXT and AVP peptides are highly similar and bind to the OXT receptor (OXTR) and the AVP receptors V1aR, V1bR and V2R (*Albers, 2015*; *Gimpl and Fahrenholz, 2001*). To investigate the specificity of OXT function in the discrimination of sex-specific cues, we characterized the sex-specific cue preference of male mice with impaired AVP signaling. Mutant mice lacking AVP die within 1 week after birth, thus preventing further analysis of adult social behaviors (*Hayashi et al., 2009*; *Russell et al., 2003*). Instead, we next attempted to investigate the potential role of AVP in sex-specific cue discrimination by ablating AVP neurons in mice generated by the cross of *Avp-Cre* mice with the ROSA26-eGFP-DTA transgenic mouse line (*Figure 2—figure supplement 1A–B*), which expresses the cellular toxin diphtheria toxin subunit A (DTA) after Cre-mediated recombination. As expected, the DTA-mediated ablation led to significant reduction in the number of AVP neurons with little effect on OXT-expressing neurons (S *Figure 2C*). Unfortunately, DTA-mediated ablation of AVP neurons induced excessive water intake and urination and grossly reduced body weight (*Figure 2—figure supplement 1C*), preventing the characterization of behavioral impairments resulting from lack of AVP.

In turn, we tested the behavior of mice lacking one or both of the two central AVP receptors, V1aR and V1bR. Three mutant mouse lines, V1aR knockout, V1bR knockout (*Wersinger et al., 2002*) and a double knockout line lacking both V1aR and V1bR, were tested for male preference toward female or male beddings. Data show that males from all three knockout lines displayed normal preference for female chemo-stimuli compared to controls (V1aR: $n_{control}$ = 10, $n_{mutant}$ = 6; V1bR: $n_{control}$ = 10, $n_{mutant}$ = 11; V1aR V1bR: $n_{control}$ = 11, $n_{mutant}$ = 11), such that both control and mutants spent more time investigating female beddings than male beddings (V1aR: $P_{control}$ <0.05, $P_{mutant}$ <0.05; V1bR: $P_{control}$ <0.001, $P_{mutant}$ <0.001; V1aR V1bR: $P_{control}$ <0.001, $P_{mutant}$ <0.001; t test) and had positive mean odor preference scores (*Figure 2A*, and *Figure 21—figure supplement 1D–I*). Thus, our results suggest that, unlike OXT, AVP is dispensable for sex-specific chemosensory preference and that the effect of OXT in sex-cue preference is mediated exclusively by OXTR.

## OXT signaling to MeA aromatase-positive neurons is required for sex discrimination

The MeA, previously shown to express moderate level of OXTR (*Yoshida et al., 2009*), is a key center for decoding socially relevant chemo-stimuli in the VNO pathway. In particular, the posterior MeA is an attractive candidate area to mediate the behavioral effects we observed in mice lacking OXT because it receives direct inputs from the accessory olfactory bulb (AOB), and displays topographically organized responses to social interactions (*Bergan et al., 2014*; *Choi et al., 2005*; *Lehman et al., 1980*). To investigate the potential role of OXT signaling in modulating the chemosensory discrimination of social cues in the posterior MeA, we genetically ablated OXTR expression from two distinct subpopulations of neurons in the posterior MeA defined by the expression of thyrotropin-releasing hormone (TRH) and aromatase, respectively, using a conditional OXTR knockout mouse line (*Oxtr^{flox/flox}*) (*Lee et al., 2008*). TRH is predominantly expressed in the posteroventral subdivision (MeApv) (Allen Mouse Brain Atlas, http://mouse.brain-map.org/gene/show/21801); while aromatase, which converts circulating testosterone to estrogen, is mainly found in neurons of the posterodorsal subdivision of the MeA (MeApd) (*Wu et al., 2009*).

To perform this experiment, we used a previously described *Trh-IRES-Cre* knock-in line (*Krashes et al., 2014*), in which Cre expression in the MeA is limited to the posteroventral subnucleus (*Figure 2—figure supplement 2A*. We also generated a *Cyp19a1(aromatase)-Cre* BAC transgenic line that faithfully recapitulates the endogenous aromatase expression in the MeApd, as well as in the other described sites of endogenous aromatase expression across the brain, such as the posteromedial BNST, preoptic hypothalamus, and lateral septum (*Balthazart et al., 1991*; *Lauber and Lichtensteiger, 1994*; *Wu et al., 2009*) (*Figure 2—figure supplement 2B,C*).

We then used the 3-chamber paradigm to examine the social interaction preference of mice lacking OXTR in subsets of neurons that express either aromatase (*Cyp19a1-Cre; Oxtr^{flox/flox}*) or TRH

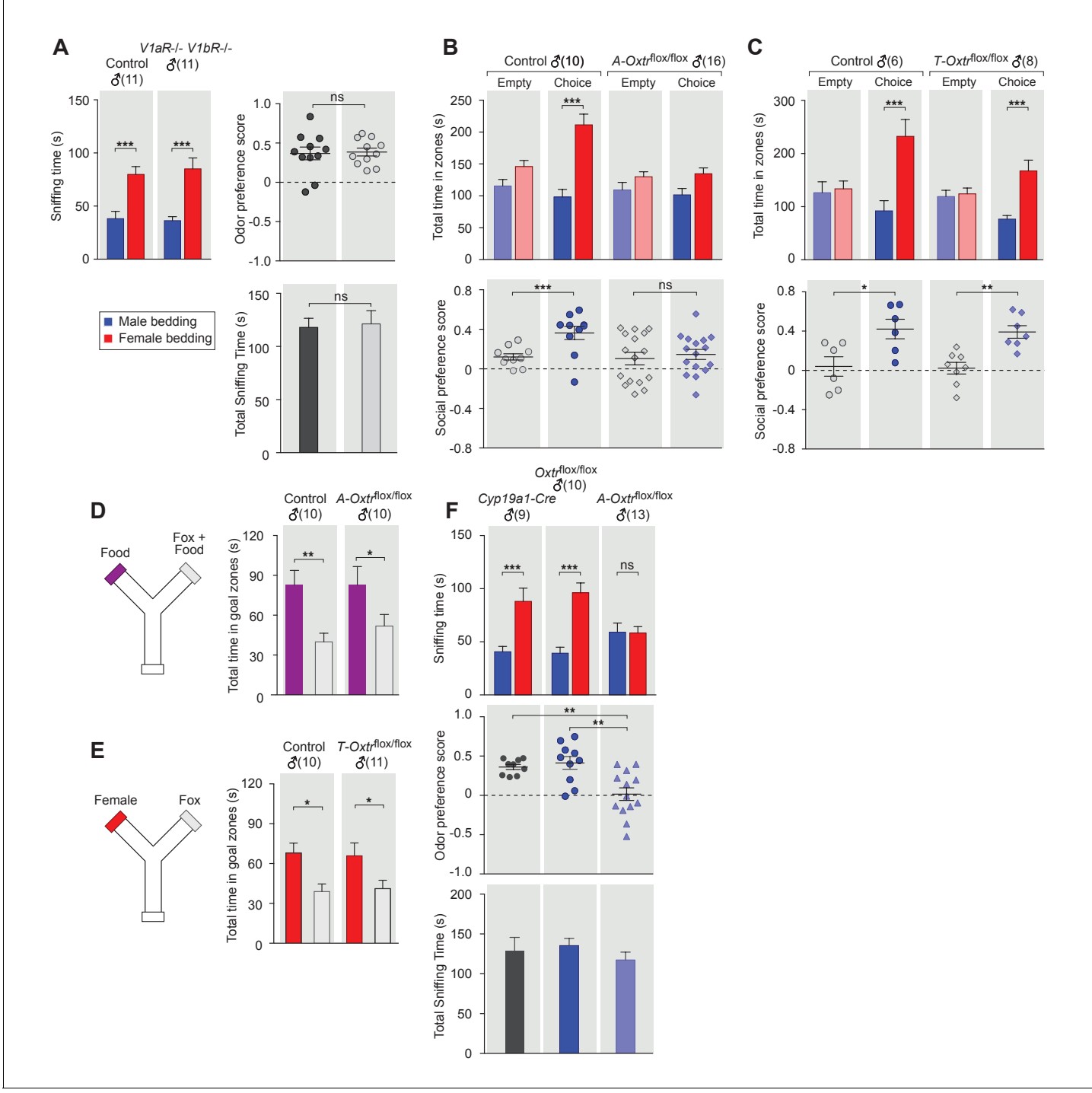

**Figure 2.** OXT signaling in aromatase-expressing neurons is required for the discrimination of male and female conspecifics. (A) Male mice lacking AVP signaling showed normal preference for female conspecific cues. Left: time spent investigating each odor cue. Upper right: odor preference score; Bottom right: total time spent in the investigation of odor cues. t test, ***p<0.001, ns, not significant. (B) Male mice lacking OXT signaling in *Cyp19a1* (aromatase)-expressing neurons (*A-Oxtr*flox/flox) have impaired preference for female conspecifics in the 3-chamber social preference test. Upper: time spent in each interaction zone. Bottom: social preference score. t test, ***p<0.001, **p<0.01, ns, not significant. (C) Comparison of control (*Oxtr*flox/flox) and *Trh-Cre; Oxtr*flox/flox (*T-Oxtr*flox/flox) mice in social interaction preference using the 3-Chamber paradigm. Upper: time spent in each interaction zone. Bottom: social preference score. t test, ***p<0.001, **p<0.01, *p<0.05. (D) Y-maze paradigm testing the willingness of control and A-*Oxtr*flox/flox male mice to investigate food vs. fox urine spiked food. t test, **p<0.01, *p<0.05. (E) Responses of control and T-*Oxtr*flox/flox mice to predator cues and female cues in a Y-maze paradigm. t test, *p<0.05. (F) Male mice lacking OXTR (A-*Oxtr*flox/flox) in aromatase-expressing neurons have impaired

*Figure 2 continued on next page*

*Figure 2 continued*
preference for female olfactory cues. Upper: time spent investigating each odor cue. Middle: odor preference score. Bottom: total time spent in the investigation of odor cues. t test, **p<0.01, ***p<0.001, ns, not significant.
DOI: https://doi.org/10.7554/eLife.31373.005
The following figure supplements are available for figure 2:
**Figure supplement 1.** Odor preference of male mice with impaired AVP signaling.
DOI: https://doi.org/10.7554/eLife.31373.006
**Figure supplement 2.** Construction of the *Cyp19a1-Cre* transgenic mouse line.
DOI: https://doi.org/10.7554/eLife.31373.007

(*Trh-Cre; Oxtr^flox/flox*). In all cases, male mice with intact OXT signaling were used as controls for littermates lacking the expression of OXTR in specific neuron types. Data show that male mice lacking OXTR expression in aromatase-positive neurons failed to show a significant preference for social interactions with female conspecifics, in a manner similar to what we observed previously with *Oxt^-/-* mice (male interaction zone = 101.5 ± 10.0 s, female interaction zone = 134.7 ± 9.1 s; not significant, t test; social preference score = 0.15 ± 0.05, preference score with empty wire cups = 0.10 ± 0.06, p=0.60, t test; n = 16; *Figure 2C*). These results suggest that OXT signaling in aromatase-expressing neurons is necessary for normal discrimination of male and female conspecific cues. In contrast, male mice with intact OXT signaling (male interaction zone = 98.2 ± 12.0 s, female interaction zone = 211.4 ± 16.9 s; p<0.001, t test; social preference score = 0.36 ± 0.07, preference score with empty wire cups = 0.12 ± 0.03; p<0.01, t test; n = 10) and male mice with impaired OXT signaling restricted to TRH neurons (male interaction zone = 76.5 ± 6.9 s, female interaction zone = 167.4 ± 20.2 s; p<0.001, t test; social preference score = 0.34 ± 0.07, preference score with empty wire cups = 0.02 ± 0.06; p<0.01, t test; n = 8) all spent significantly longer durations in the female interaction zone and had a strong preference for female conspecifics (*Figure 2C* and *Figure 2D*).

Next, we tested the various mutant lines and littermate controls in bedding investigation assays. Normal preference in investigating female odor was observed in *Oxtr^flox/flox* (male bedding = 39.3 ± 5.6 s, female bedding = 96.1 ± 9.2 s; p<0.001, multiple t-tests; preference score = 0.41 ± 0.08; n = 10) and *Cyp19a1-Cre* (male bedding = 40.7 ± 4.9 s, female bedding = 88.1 ± 12.5 s; p<0.001, t test; preference score = 0.36 ± 0.03, n = 9) male mice, both of which have intact OXT signaling. In contrast, the homozygous offspring of these two lines, which have impaired OXT signaling in aromatase-positive neurons, showed loss of preference to investigate female bedding (n = 13; male bedding = 59.0 ± 8.7 s, female bedding = 58.4 ± 5.9 s; p=0.95, t test; preference score = 0.02 ± 0.08, significantly lower than that of *Oxtr^flox/flox* and *Cyp19a1-Cre*, p<0.05, t test; *Figure 2G*). We also utilized the Y-maze to measure the willingness of mice lacking OXTR expression in specific neuron types to explore attractive cues versus predator cues. Consistent with the apparently normal ability of *Oxt^-/-* male mice to detect predator cues, male mice with impaired OXT signaling in aromatase-positive (Food zone = 82.7 ± 14.0 s, Food +fox urine zone = 51.8 ± 8.8 s; p<0.05, t test; n = 10) or TRH positive neurons (Female chamber = 65.8 ± 9.7 s, Fox urine zone = 41.1 ± 6.3 s; p<0.05, t test; n = 11) still spent significantly longer durations in the goal chamber with the attractive cue of either food or female urine than the goal chamber containing fox urine (*Figure 2E* and *Figure 2F*).

Taken together, our results demonstrate that aromatase-positive neurons represent a critical site for OXT action in the control of chemosensory discrimination of male versus female conspecifics. Yet, the ablation of OXTR from aromatase-positive neurons using the *Cyp19a1-Cre* line also affects other sites expressing this gene. Of particular interest are the aromatase-expressing neurons of the bed nucleus of the stria terminalis (BNST), another first order target of neurons in the AOB, and a key region critical for the processing the chemosensory information (*Figure 2—figure supplement 2C*). To determine whether the function of OXT signaling in chemosensory discrimination of conspecifics is restricted to aromatase-positive neurons in the MeA, we used a virus-assisted strategy to selectively abolish OXT signaling in specific brain areas of adult mice. Adult *Oxtr^flox/flox* male mice were stereotaxically injected with a recombinant AAV virus with CMV-driven expression of a GFP-Cre fusion protein (AAV-GFP-Cre) in the bilateral BNST or MeA (*Figure 3A*). Mice receiving similar injection of an AAV virus with CMV driven expression of GFP (AAV-GFP) served as controls. The

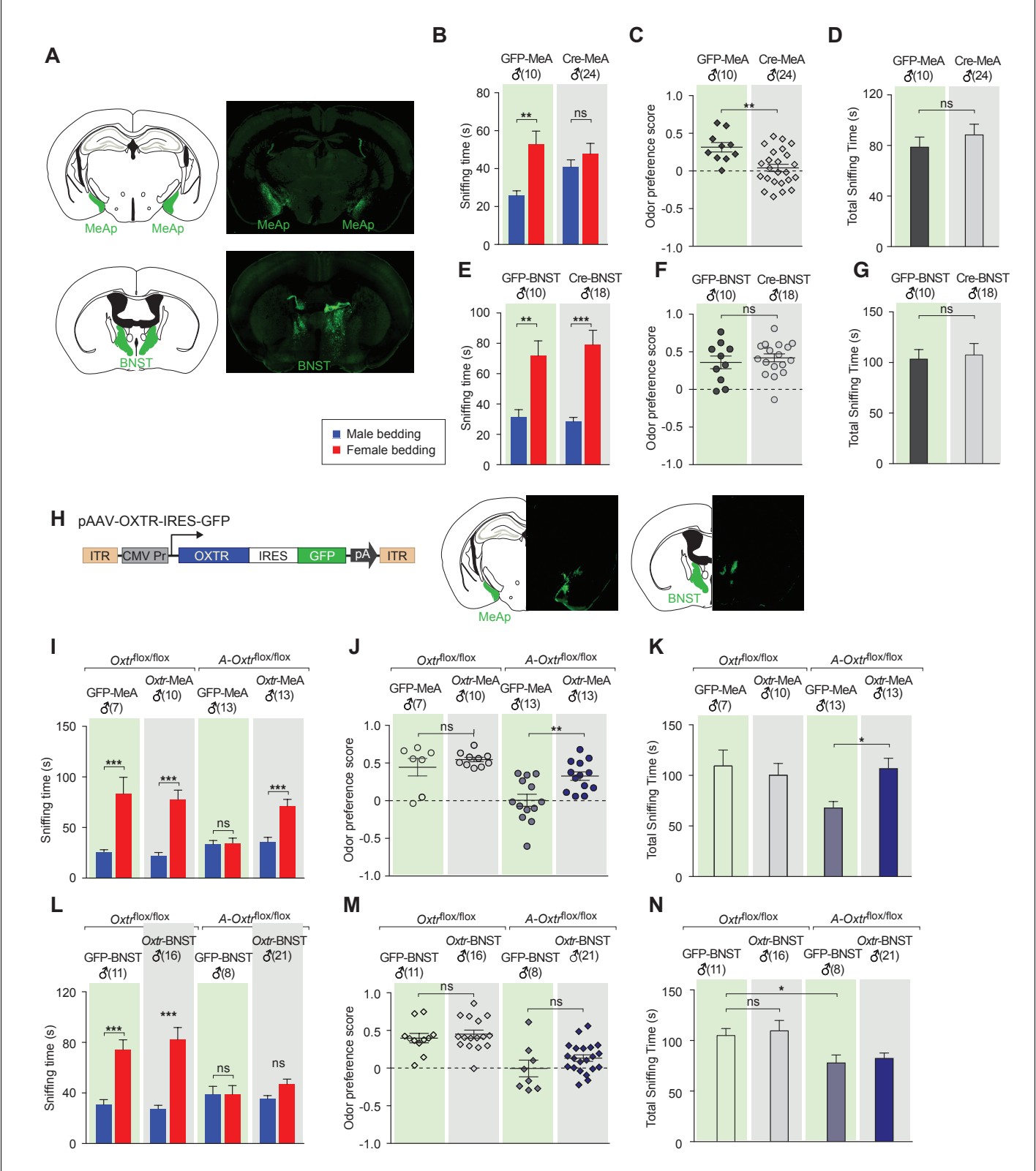

**Figure 3.** Aromatase-expressing neurons in the MeApd are the cellular substrate mediating OT signaling in sexual discrimination. (**A**) Virus-mediated ablation of OXTR in the MeA (Upper) and BNST (Bottom) of *Oxtr*^flox/flox^ mice. Illustrations of the MeAp and BNST were adapted from the Paxinos and Franklin mouse brain atlas. The AAV-eGFP and AAV-Cre-eGFP viruses are stereotaxically injected into the MeAp and BNST of *Oxtr*^flox/flox^ male. (**B–D**) Time spent investigating each odor cue, odor preference scores, and total time investigating odor cues of *Oxtr*^flox/flox^ mice with either AAV-eGFP or

*Figure 3 continued*

AAV-Cre-eGFP injection of the MeAp. t test, **p<0.01, ns, not significant. (**E–G**) Effects of OXTR ablation in the BNST on time investigating each odor cue, odor preference scores, and total time investigating odor cues. t test, ***p<0.001, **p<0.01, ns, not significant. (**H**) Virus-mediated expression of OXTR in the MeA and BNST via the AAV-OXTR-IRES-eGFP virus. (**I–N**) Time spent investigating each odor cue, odor preference scores, and total time investigating odor cues of *Oxtr^flox/flox* (control) and *Cyp19a1-Cre; Oxtr^flox/flox* (*A-Oxtr^flox/flox*) mice with MeAp (**I–K**) or BNST (**L–N**) virus injection. t test, ***p<0.001, **p<0.01, *p<0.05.

DOI: https://doi.org/10.7554/eLife.31373.008

specificity of targeting was verified by examining GFP expression along the rostrocaudal axis of virally injected brains. No diffusion of virus between the two targets was observed. Using the chemo-stimuli-preference paradigm, we tested the ability of male mice with targeted manipulation of OXTR expression to discriminate between male and female conspecific cues. We found that mice with specific impairment of OXT signaling in the MeA displayed loss of preference for female chemostimuli (male bedding = 40.8 ± 3.8 s, female bedding = 47.6 ± 5.7 s; p=0.29, multiple t-tests; preference score = 0.04 ± 0.05, n = 24), whereas control (GFP in MeA: male bedding = 25.7 ± 2.6 s, female bedding = 52.9 ± 6.8 s; p<0.01, t test; preference score = 0.32 ± 0.06, n = 10; GFP in BNST: male bedding = 31.6 ± 4.7 s, female bedding = 71.7 ± 9.9 s; p<0.01, t test; preference score = 0.36 ± 0.08, n = 10) and male mice with impaired OXT signaling in the BNST (GFP in MeA: male bedding = 28.4 ± 2.8 s, female bedding = 79.0 ± 9.6 s; p<0.001, t test; preference score = 0.42 ± 0.05, n = 18) presented normal preference in investigating female bedding (*Figure 3B–G*). The odor preference score of male mice with impaired OXT signaling in the MeA was significantly lower than that of control mice with GFP expression in the MeA (p<0.01, t test), whereas impaired OXT signaling in the BNST had no effect on the odor preference (p=0.52, t test). Thus, our results suggest that aromatase-positive neurons of the MeA, but not of the BNST, mediate the effect of OXT on the discrimination of sex-specific cues by male mice.

To provide further evidence for the specific role of MeA aromatase-positive neurons in sex-cue discrimination, we performed a series of experiments in which OXTR expression was restored in either the MeA or BNST of *Cyp19a1-Cre; Oxtr^flox/flox* male mice. A recombinant AAV virus in which the CMV promoter drives constitutive expression of OXTR (AAV-OXTR) or a control virus with CMV-driven expression of GFP was stereotaxically delivered (*Figure 3H*) and behaviors were measured using the chemostimuli preference paradigm (*Figure 3I–N*). We found that rescue of OXTR expression in the MeA, but not the BNST, was sufficient to restore the preference for female chemostimuli in male mice. Bilateral injection of AAV-OXTR to the MeA led to increased investigation of female bedding in mutant mice (male bedding = 35.7 ± 4.6 s, female bedding = 71.0 ± 6.9 s; p<0.01; t test; preference score = 0.33 ± 0.06, n = 13; *Figure 3I–K*), whereas virus-mediated OXTR expression in the BNST had no effect (male bedding = 35.3 ± 2.6 s, female bedding = 46.9 ± 4.0 s; p>0.05; t test; preference score = 0.13 ± 0.04, n = 21; *Figure 3L–N*).

In summary, our data indicate that OXT signaling is required in MeApd aromatase-positive neurons for the discrimination of sex-specific cues. This role may rely on one or two of the following mechanisms. OXT may have a developmental role on the aromatase-positive neurons of the MeA to ensure the proper configuration of this neuronal population to control sex discrimination. Alternatively, or in addition, OXT signaling may be acutely required in adults to modulate sex discrimination. OXT release is under dynamic regulation in adults, and can be triggered by sensory stimulations and psychosocial stressors (*Landgraf and Neumann, 2004*; *Veenema and Neumann, 2008*), consistent with the possibility that OXT is capable of acutely modulating the function of neural circuitries in adults in response to sensory cues. Although our results cannot rule out a possible role of OXT in configuring neural circuits regulating sex discrimination during development, the virus-mediated ablation and rescue of OXTR expression in adult MeA lead to corresponding changes in the ability of male mice to discriminate female and male conspecific cues, thus strongly suggesting that defects in chemosensory discrimination are a direct consequence of acute impairment of OXT signaling in the adult MeA.

## Loss of OXT signaling leads to altered chemosensory response profiles in the MeA

The MeA relays chemosensory inputs from the VNO pathway to hypothalamic nuclei controlling behavioral and endocrine responses, thus playing a key role in the regulation of social behaviors. To investigate how OXT signaling in the MeA alters sensory responses to social stimuli on a moment-to-moment basis, we used multisite extracellular recording and monitored neuronal activation patterns in the MeA upon odor stimulation in the presence or absence of OXT signaling (*Bergan et al., 2014*). Multisite extracellular recordings of the MeA were conducted in anesthetized wild-type and mutant mice, with the VNO pump activated by electrical stimulation of the sympathetic nerve trunk to allow nonvolatile stimuli, including female urine, male urine and predator urine, to access the VNO neuroepithelium (*Figure 4*; *Ben-Shaul et al., 2010*). Each sensory stimulus was presented with 6–12 randomly interleaved repetitions for all experiments.

Electrophysiological probes consisted of 32 recording sites evenly distributed dorso-ventrally over 1.55 mm (50 µm between sites; Neuronexus). Identification of single unit activity was based on spike shape, clustering of principal component projections, and a clear refractory period between successive spikes (*Harris et al., 2000*; *Hazan et al., 2006*). Consistent with MeA anatomy, units responsive to VNO sensory stimuli were typically found from 0 to ~1.4 mm from ventral surface of the brain. Electrode probes were dipped in a fluorescent dye, and accurate targeting of probes was confirmed with postmortem histology (*Figure 4C*).

Sensory-evoked responses were identified by comparing the spike rates before stimulus presentation (20 s prior to stimulus presentation) to spike rates following stimulus presentation (40 s following stimulus presentation) using a non-parametric ANOVA. Units were considered 'responsive' if any stimulus elicited a p-value less than 0.01. MeA neurons from control *Oxtr^{flox/flox}* male mice were compared to that of mutant male mice lacking OXTR signaling in aromatase-positive neurons. A total of 106 single units were recorded from the MeA of control male mice (9 animals; 10–13 weeks old), with 40 units responding to VNO cues with a p value <= 0.01 (non-parametric ANOVA) and 191 single units were recorded from the MeA of mutant male mice (13 animals; 10–13 weeks old), with 70 units responding to VNO cues with a p value <= 0.01 (non-parametric ANOVA).

Single units responsive to VNO cues were categorized by the sensory stimulus that elicited the strongest response: female, male or predator (*Figure 4D*). In accord with previous c-fos and electrophysiological findings (*Bergan et al., 2014*; *Choi et al., 2005*; *Kang et al., 2011*; *Samuelsen and Meredith, 2009*), MeA units in male mice with wild-type OXTR function were most responsive to female (23.6%) and predator (13.2%) cues, while less than 1% of MeA units from control males responded to male stimuli (*Figure 4D–G*). By contrast, in male mice lacking OXT signaling in the aromatase-expressing neurons, the proportion of MeA units responsive to female stimuli was reduced (6.2%), while the proportions of units responsive to male (6.7%) and predator (23.3%) stimuli were increased compared to control male mice (*Figure 4D*). The observed difference in response rates between control mice and male mice lacking OXT signaling in the aromatase-expressing neurons was significant for each stimulus (female: p<=0.0001; male: p<=0.02; predator: p<=0.02; permutation test). We found that the response strength (spikes/second) to conspecific cues was not dramatically changed by permanent loss of OXT signaling although the average response strength of predator neurons was reduced by ~10% in mutant male mice as compared to control male mice (*Figure 4E*; p<0.05). These results suggest that the inability to discriminate sex-specific cues observed in mice with impaired OXT signaling (*Figure 2*) coincides with a clear reorganization of sensory responses to social stimuli in the MeA.

## Repression of OXT signaling acutely modulates conspecific social interaction and sensory response profiles

To further elucidate the temporal requirement of OXT signaling in the chemosensory discrimination of female and male cues, we transiently suppressed OXT signaling by intraperitoneal injection of OXTR antagonist (L-368,899 hydrochloride) at a dosage of 5 mg/kg$^{-1}$ (*Dölen et al., 2013*; *Nakajima et al., 2014*), followed by behavioral analysis (*Figure 5A–C*) and electrophysiological recording (*Figure 5D–J*). We found that OXTR antagonist administration specifically reduced the time that male mice spent in the female interaction zone (*Figure 5A–B*), without affecting the time spent in the male interaction zone or the total distance travelled (saline: female interaction

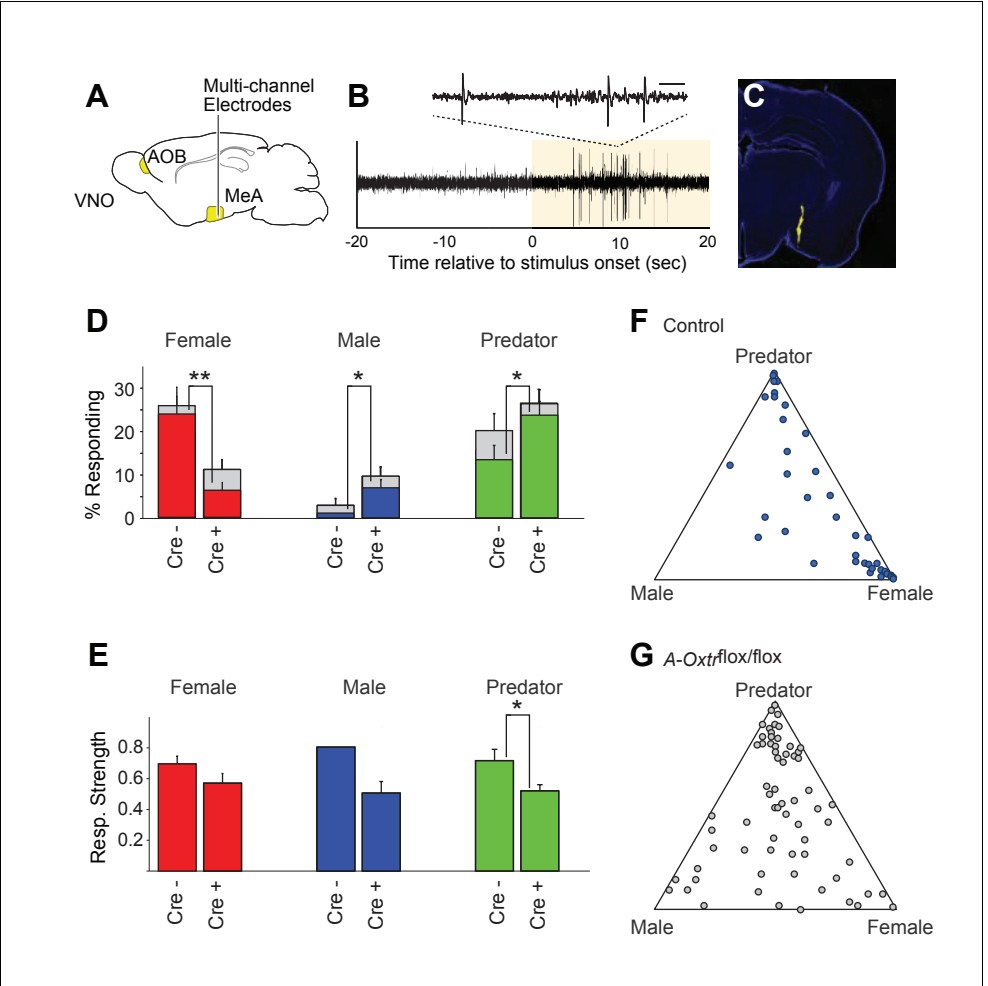

**Figure 4.** OXT signaling is critical for dimorphic neuronal responses of the MeA to conspecific cues. (**A**) Schematic showing core elements of the vomeronasal pathway (yellow) and the targeting of multichannel probes to the MeA. (**B**) Single voltage trace showing an increase in the activity of a well-isolated unit following VNO sensory stimulation (yellow). Inset shows a subset of the voltage trace at higher time resolution. (**C**) DiI labeled electrode tract located squarely in the MeA (blue: DAPI; Yellow: DiI). (**D**) Comparison of responses to male, female, and predator stimuli in *Oxtr^flox/flox* and *Cyp19a1-Cre; Oxtr^flox/flox* male mice. Grey bars: percentage of single MeA units that responded to each stimulus (nonparametric ANOVA; p<=0.05). Colored bars indicate the percentage of single units that responded most strongly for the indicated stimulus. Asterisks indicate a significant difference in response rate between *Oxtr^flox/flox* and *Cyp19a1-Cre; Oxtr^flox/flox* animals (permutation t-test; *p<0.05; **p<0.0001). (**E**) Comparison of response strength to male, female, and predator stimuli in *Oxtr^flox/flox* and *Cyp19a1-Cre; Oxtr^flox/flox* (A-Oxtr^flox/flox) animals. Response strength = (post-pre)/(post +pre); where for a given neuron post = the firing rate in 40 s after the stimulus was presented and pre = the firing rate in 20 s prior to when the stimulus was presented. Asterisks indicate a significant difference in response rate between *Oxtr^flox/flox* and *A-Oxtr^flox/flox* animals (permutation t-test; *p<0.01) (**F-G**) Selectivity of MeA units to sensory stimuli in control (*Oxtr^flox/flox*) (**F**) and *A-Oxtr^flox/flox* (**G**) animals. Each point represents the sensory responses of an individual unit with at least one significant response to male, female, or predator stimuli. Points located near a vertex indicate selective responses for the indicated stimulus. Points located in the center of the plot indicate less selective responses.
DOI: https://doi.org/10.7554/eLife.31373.009

zone = 159.8 ± 18.1 s, male interaction zone = 93.8 ± 8.9 s; p<0.001; preference score = 0.24 ± 0.04, total distance travelled = 3709 ± 160 cm; OXTR antagonist: female interaction zone = 106.4 ± 11.0 s, male interaction zone = 101.6 ± 12.7 s; p=0.79; preference score = 0.04 ± 0.08, total distance travelled = 3360 ± 247 cm; n = 12; *Figure 5C*). The changes at the behavioral level could be observed 30 min after IP injection, demonstrating that normal social

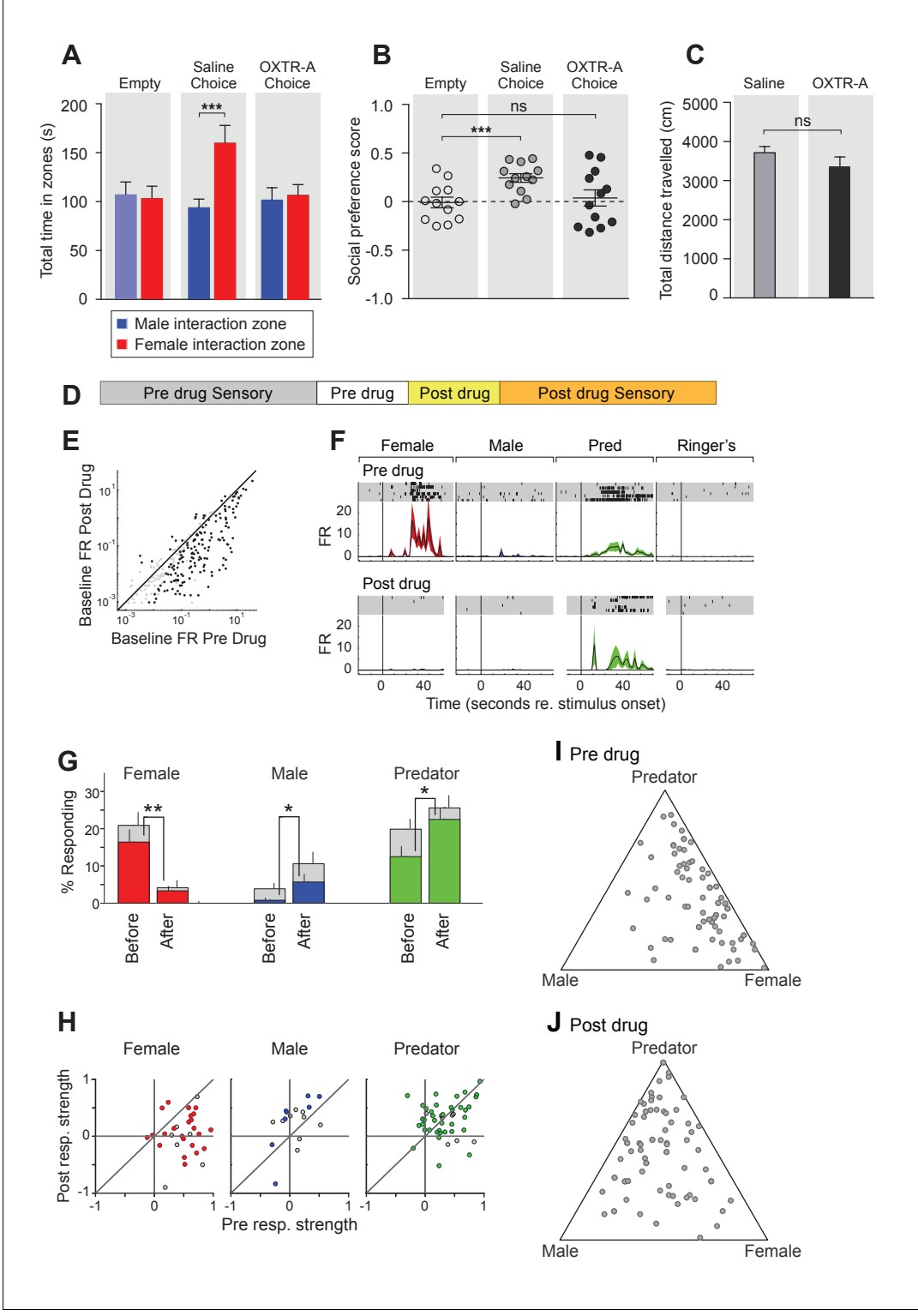

**Figure 5.** Acute pharmacological inhibition of OXTR leads to impaired social interaction preference and altered neuronal response profile in the MeA. (**A–C**) Comparison of times spent in each interaction zone (t test), social preference scores (paired t test) and distance traveled (paired t test) of the subjects (*Oxtr*$^{flox/flox}$) before and after IP injection of OXTR antagonist (OXTR-A). ***p<0.001, **p<0.01. (**D**) Structure of combined pharmacology and electrophysiology experiments. Sensory responses were determined prior to drug application (~50 min), followed by a 10 min baseline period. Drugs were injected IP, and a second baseline period was collected. Finally, sensory

*Figure 5 continued on next page*

*Figure 5 continued*

responses were determined after drug application. (**E**) Firing rate (spikes/second) for single units in the absence of sensory stimuli (abscissa: firing rate before antagonist; ordinate: firing rate after antagonist; log scale). (**F**) Responses of a single MeA unit to sensory stimuli before and after the mouse was injected with OXTR-A. For each panel, raster plots indicating the timing of individual action potentials elicited by multiple presentations of the same stimulus (shaded region). Histograms of the mean response and standard error for the same data are shown below each raster plot. Time zero indicates alignment to the start of stimulus presentation. (**G**) Comparison of responses to male, female, and predator stimuli before and after antagonist injection. Grey bars: percentage of single MeA units that responded to each stimulus (nonparametric ANOVA; $p \leq 0.05$). Colored bars: percentage of single units that responded most strongly for the indicated stimulus. Asterisks indicate a significant difference in the fraction of responsive units before and after OXTR-A injection (permutation t-test; *$p<0.05$; **$p<0.001$) (**H**) Comparison of response strength before and after OXTR-A injection for all single units responding to a given stimulus. Grey points indicate single units with a statistically significant response to the stimulus plotted in each axis. Colored points indicate units with a significant response that was also the strongest response (compared with responses to the other stimuli). Abscissa: (post-pre)/(post + pre) prior to OXTR-A injection. Ordinate: (post-pre) / (post + pre) after OTRA injection. (**I–J**) Selectivity of MeA units to sensory stimuli before (**I**) and after (**J**) OXTR-A injection. Each point represents the sensory responses of an individual unit with at least one significant response to male, female, or predator stimuli.

DOI: https://doi.org/10.7554/eLife.31373.010

The following figure supplement is available for figure 5:

**Figure supplement 1.** Mapping of OXT outputs from the PVN and the SON.

DOI: https://doi.org/10.7554/eLife.31373.011

interactions with conspecifics rely on OXT signaling in adults and that acute modulation in OXT signaling can promptly modify behavioral responses.

To assess the acute function of OXT signaling at the sensory response level, we compared the responses of single MeA neurons before and after systemic OXTR antagonist injection (***Figure 5D–J***). Before pharmacological manipulation, responses of MeA units to repeated sensory stimulation were determined as described previously. Neurons were monitored without sensory stimulation for 10 min before and 10 min after OXTR antagonist IP injection to identify a direct effect of the compound on neural activity. Data show that, in the absence of sensory stimulation, injection of the OXTR antagonist decreased the firing rate of MeA units (***Figure 5E***). Sensory responses were then reassessed in the presence of OXTR antagonist (***Figure 5D***). We restricted our analyses to single MeA units that were maintained before, during, and after antagonist injection. A total of 129 single units from the MeA of *Oxtr^{flox/flox}* male mice were maintained for the entire experiment, which typically lasted 2 hr (11 animals; 10–13 weeks old).

Data show that OXTR antagonist reduced the number of neuronal responses categorized as 'female' (***Figure 5G***; 16.4% responding before; 3.3% responding after; $p \leq 0.001$, permutation test), increased the number of neuronal responses categorized as 'male' (0.8% responding before; 5.7% responding after; $p \leq 0.026$, permutation test), and increased the number of neuronal responses categorized as 'predator' (0.8% responding before; 5.7% responding after; $p \leq 0.026$, permutation test). Because units were classified based only on the stimulus that elicited the strongest response, these categorization changes could reflect a reduction in female responses and increase in male and predator responses, or both. A single MeA unit, showing a particularly dramatic antagonist mediated change in sensory responses, is shown in ***Figure 5F***. Prior to drug injection this unit responded approximately evenly to both predator and female stimuli. Following drug injection, the unit responded nearly exclusively to predator stimuli.

Next, we compared the response strengths for all units that responded in either the pre-drug or post-drug epoch (see Materials and methods). OXTR antagonist application dramatically reduced the strength of female-responsive units ($p=0.003$; Wilcoxon signed rank test; ***Figure 5H***, red), modestly increased the strength of male-responsive units ($p=0.07$; Wilcoxon signed rank test; ***Figure 5H***, blue), and showed no clear effect on predator responsive units ($p=1$; Wilcoxon signed rank test; ***Figure 5H***, green). This suggests that the OXT-dependent changes in MeA sensory responses persists into adulthood, and that the neural circuits regulating social discrimination can be manipulated acutely by repressing OXT signaling (***Figure 5I,J***). Moreover, in view of the decrease in response

strength to female stimuli following OXTR antagonist (*Figure 5H*), but not males or predators, it is clear that the reduced response to female stimuli is the major effect of OXTR antagonist in the MeA.

To further confirm the critical role of OXT-mediated acute neuromodulation in the discrimination of female and male conspecifics, we investigated the effects of direct manipulation of OXT-expressing neurons in conspecific social interaction. OXT-expressing neurons are mainly distributed in two hypothalamic areas, the paraventricular nucleus (PVN) and the supraoptic nucleus (SON). To systematically map the distribution and origins of OXT fibers from the two populations of OXT-expressing neurons, a Cre-dependent AAV-ChR2-YFP virus that readily enables visualization of neuronal fibers was stereotaxically delivered to either the PVN or the SON of an OXT-iCre line (2 female and 2 male virgin mice for each target site; *Figure 5—figure supplement 1*) (*Gradinaru et al., 2009*; *Wu et al., 2012*), and fine OXT fibers could be observed after GFP immunostaining. The distribution of OXT fibers did not exhibit any obvious sexual dimorphism between virgin female and male mice, and female and male results were combined when analyzing the fiber distribution patterns from OXT-expressing neurons in the PVN or the SON. Consistent with previous reports (*Buijs et al., 1983*; *Knobloch et al., 2012*), our results show that OXT-expressing neurons in the PVN, but not those in the SON, are the major contributor of OXT fibers in all 27 brain areas examined (*Figure 5—figure supplement 1*). Medium to high levels of OXT fibers originated from the PVN can be found in the BNST and MeA, two chemosensory nuclei with dense populations of aromatase-expressing neurons, whereas only sparse OXT fibers from the SON were found in the two areas. Although OXT can be released from dendrites and somas, local axonal release of OXT is suggested to specifically control region-associated behaviors (*Knobloch et al., 2012*). OXT-expressing neurons in the PVN, which are the major contributor of OXT fibers in the MeA, may also be the major provider of OXT peptides modulating neuronal responses of MeApd neurons to chemostimuli.

To examine the effect of reversible modulation of OXT peptide levels on a longer time scale, we aimed to directly control the activities of OXT-expressing neurons by using the Designer Receptor Exclusively Activated by Designer Drugs (DREADD). Cre-dependent AAV virus conditionally expressing the inhibitory DREADD (AAV8-hSyn-DIO-hM4Di-mCherry) was stereotaxically delivered either to the PVN (*Figure 6A,B,D,E,H,I,L and M*) or the SON (*Figure 6B,C,F,G,J,K,N and O*) of the *Oxt-Cre* line (*Figure 6—figure supplement 1*) (*Armbruster et al., 2007*; *Krashes et al., 2011*; *Krashes et al., 2014*; *Wu et al., 2012*). After binding of the designer drug clozapine-N-oxide (CNO), the inhibitory DREADD reduces neuronal activities via the Gi pathway, therefore preventing the release of OXT peptides. We assessed the effects of chemogenetic inhibition of OXT-expressing neurons in social interactions by comparing the preference of virally infected male mice after IP injection of saline and at various intervals after IP injection of CNO. As expected, male mice with saline injection showed a strong preference for investigating a female mouse over a male mouse in the 3-chamber paradigm (*Figure 6D–G*). After a single injection of CNO, OXT-iCre mice with inhibitory DREADD virus infection in the PVN presented reduced preference in investigating female conspecifics 1 hr after drug injection (habituation: empty male interaction zone = 121.2 ± 11.8 s, empty female interaction zone = 119.1 ± 8.0 s; p=0.92, t test; preference score with empty wire cups = 0.01 ± 0.04; saline: male interaction zone = 90.2 ± 12.2 s, female interaction zone = 171.5 ± 21.2 s; p<0.001, t test; social preference score = 0.31 ± 0.06, significantly higher than that with empty wire cups, p<0.001, paired t test; CNO: male interaction zone = 108.2 ± 13.3 s, female interaction zone = 114.7 ± 18.9 s; p=0.76, t test; social preference score = −0.002 ± 0.08, not significantly differing from that with empty wire cups, p=0.92, paired t test; n = 10; *Figure 6D–E*).

This phenomenon of impaired preference could still be observed 2 days after CNO injection (habituation: empty male interaction zone = 121.5 ± 10.0 s, empty female interaction zone = 116.3 ± 7.2 s; p=0.76, t test; preference score with empty wire cups = −0.02 ± 0.03; Choice: male interaction zone = 145.6 ± 13.7 s, female interaction zone = 142.6 ± 15.6 s; p=0.86; social preference score = −0.02 ± 0.05, not significantly differing from that that with empty wire cups, paired t test; n = 10; *Figure 6H–I*), while mice appeared to regain normal preference within 1 week of drug injection (habituation: empty male interaction zone = 130.1 ± 10.1 s, empty female interaction zone = 121.70.1±9.4 s; p=0.68, t test; preference score with empty wire cups = −0.03 ± 0.03; choice: male interaction zone = 111.8 ± 13.7 s, female interaction zone = 187.3 ± 20.8 s; p<0.001, t test; social preference score = 0.25 ± 0.05, significantly higher than that with empty wire cups, p<0.001,

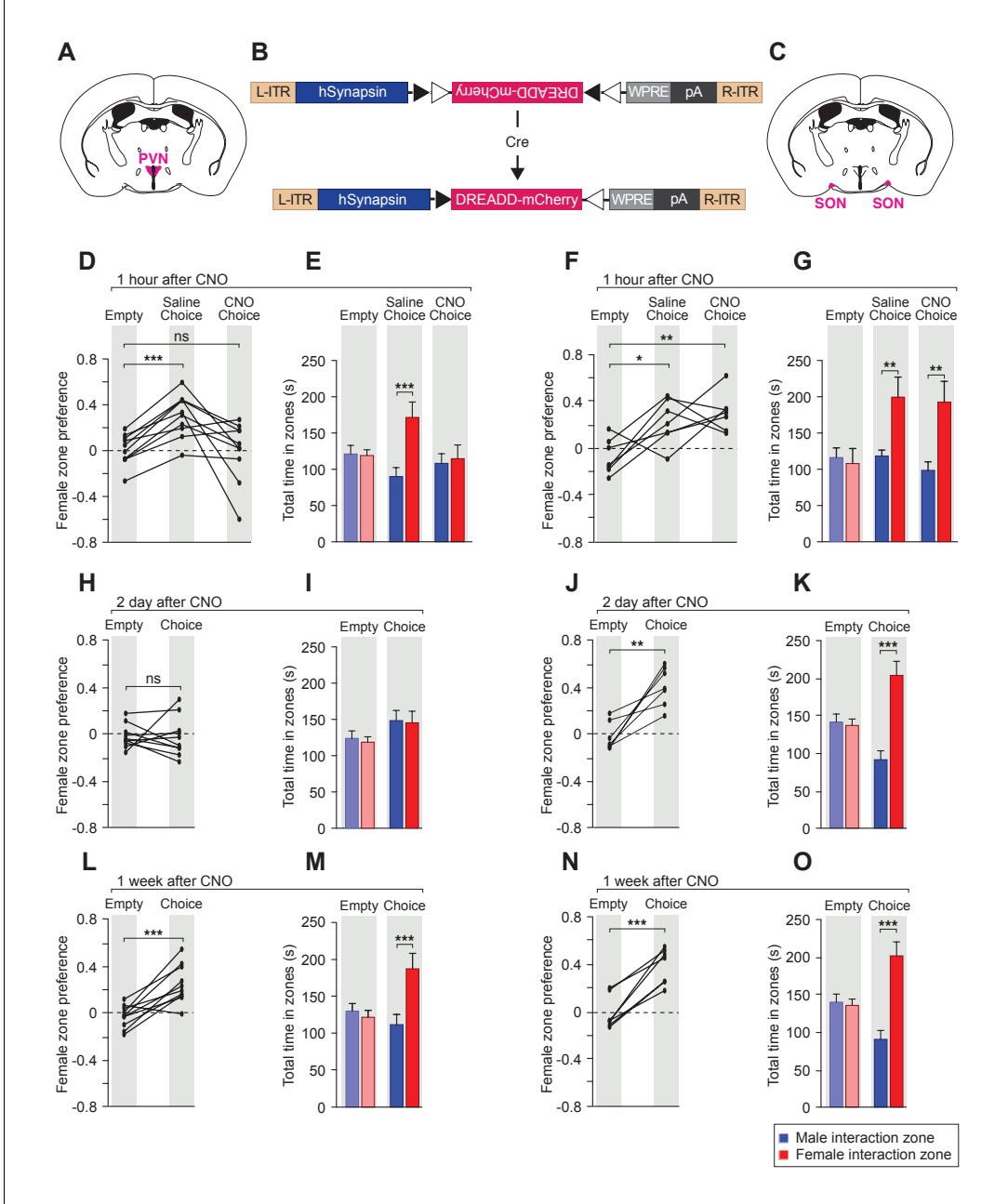

**Figure 6.** Acute chemogenetic inhibition of OXT signaling is sufficient to disturb the discrimination of female and male conspecifics. (A–C) Schematic of hM4Di mediated inhibition of OXT-expressing neurons. Illustration of the PVN (A) and SON (C) in coronal sections are adapted from the Paxinos and Franklin mouse brain atlas. The AAV8-hSyn-DIO-hM4Di-mCherry virus (B) is stereotaxically injected into either the PVN or SON of *Oxt-IRES-Cre* (*Oxt-iCre*) male mice. Rapid and reversible inhibition of the OXT-expressing neurons can be induced by IP injection of CNO. (D, E, H, I, L, M) Time courses of changes in the social investigation preference scores (paired t test) and durations that the subjects spent in each sniffing zone (t test) of *Oxt-iCre* male mice after CNO-mediated inhibition of OXT-expressing neurons in the PVN. ***p<0.001 (F, G, J, K, N, O) Time courses of changes in the social investigation preference scores (paired t test) and durations that the subjects spent in each sniffing zone (t test) of *Oxt-iCre* male mice after CNO-mediated inhibition of OXT-expressing neurons in the SON. ***p<0.001, **p<0.01, *p<0.05.

DOI: https://doi.org/10.7554/eLife.31373.012

The following figure supplement is available for figure 6:

**Figure supplement 1.** DREADD AAV virus-mediated manipulation of OT neurons.
DOI: https://doi.org/10.7554/eLife.31373.013

paired t test; n = 10; *Figure 6L–M*). Thus, direct modulation of the neuronal source of OXT in the PVN can lead to fast and reversible changes in social interaction preference.

Consistent with the PVN being the main origin of OXT fibers in the MeA, OXT-iCre mice with DREADD virus infection in the SON did not show any significant change in social interaction after CNO injection (differences in time spent in interaction zones tested via the t test, and differences in social preference scores tested via paired t test; ***p<0.001, **p<0.01, *p<0.05; n = 7; *Figure 6F– G, J–K and N–O*).

Two sets of controls confirmed that the observed DREADD-mediated effects of CNO were specific to the activation or inhibition of OXT-expressing neurons. First, the behavior of a given mouse was compared before and after CNO injection. Prior to CNO injection, *Oxt-iCre* male mice with DREADD virus infection in either PVN or SON presented normal preference for female mice, excluding the possibility of ligand-independent effect of DREADDs. Second, in contrast to the effects observed post CNO injection with *Oxt-iCre* male mice expressing the inhibitory DREADD in the PVN, *Oxt-iCre* male mice expressing the inhibitory DREADD in the SON presented normal preference towards female mice before and after CNO injection, excluding the possibility that the impaired social preference is due to an off-target effect of CNO or its metabolite clozapine (*Gomez et al., 2017*; *Saloman et al., 2016*).

## Enhanced OXT release acutely modulates behavior and sensory response profiles

As a complement to pharmacological and genetic manipulations inhibiting OXT signaling, we explored whether enhancing the activity of OXT-expressing neurons in the PVN would alter sexual preference behavior and electrophysiological response profiles of neurons in the MeA of males. To achieve this, we targeted the PVN of *Oxt-iCre* mice with excitatory DREADD (AAV8-hSyn-DIO-hM3Dq-mCherry) to allow the activation of PVN OXT-expressing neurons by IP injection of CNO (*Figure 6—figure supplement 1*). Elevated central OXT levels are known to induce self-grooming of the facial, truncal, and genital areas in rodents (*Amico et al., 2004*; *Caldwell et al., 1986*; *Drago et al., 1986*), which can be blocked by OXTR antagonists (*Amico et al., 2004*). Indeed, IP injection of CNO caused intense self-grooming in *Oxt-iCre* mice expressing excitatory DREADD in the PVN OXT-expressing neurons, indicating acute elevation of central OXT signaling. This intense grooming behavior starting minutes after CNO injection was seen to gradually diminish only after about 15 min, impairing our ability to detect the immediate changes in social preference behaviors triggered by acute elevation of OXT levels.

When tested 15 min after CNO injection, that is after the phase of intense grooming had subsided, *Oxt-iCre* mice expressing excitatory DREADD in the PVN showed a loss preference for female stimuli (*Figure 7—figure supplement 1A*), which did not recover 2 days, 1 week and even 2 weeks post CNO injection (*Figure 7—figure supplement 1B–C*).

To investigate the impact of acute increase in OXT levels on neuronal responses to sensory stimuli, we profiled the responses of single units in the MeA of male *Oxt-iCre* mice injected with excitatory DREADDs in the PVN before and after CNO injection (*Figure 7D*). CNO modestly increased the firing rate of MeA units in the absence of sensory stimulation (*Figure 7E*). A total of 98 single units were recorded from the MeA to compare the effect of chemogenetic activation of OXT-expressing neurons in the PVN. A single MeA unit, illustrating an increase in responses to female stimuli is shown in *Figure 7F*. Though several spikes are evident prior to CNO injection, this unit was relatively unresponsive. Following CNO injection, a strong response to female stimuli is revealed, demonstrating that increased response strength to female stimuli can be observed in units that were almost entirely silent prior to DREADD-mediated activation (*Figure 7F*). We found that the OXT activation by CNO increased the number of neuronal responses categorized as 'female' (*Figure 7G*; 16.9% responding before; 29.3% responding after; p<=0.03, permutation test), modestly decreased the number of neuronal responses categorized as 'male' (3.8% responding before; 0.0% responding after; p<=0.069, permutation test), and had little effect on the number of neurons categorized as predator (12.3% responding before; 10.3%).

We next compared the response strengths for all units that responded to sensory stimuli before or after CNO injection. CNO injection increased the strength of female-responsive units (p=0.0005; Wilcoxon signed rank test; *Figure 7H*, red), while having little effect on the response strength of male (p=0.63; Wilcoxon signed rank test) or predator (p=0.82; Wilcoxon signed rank test) evoked

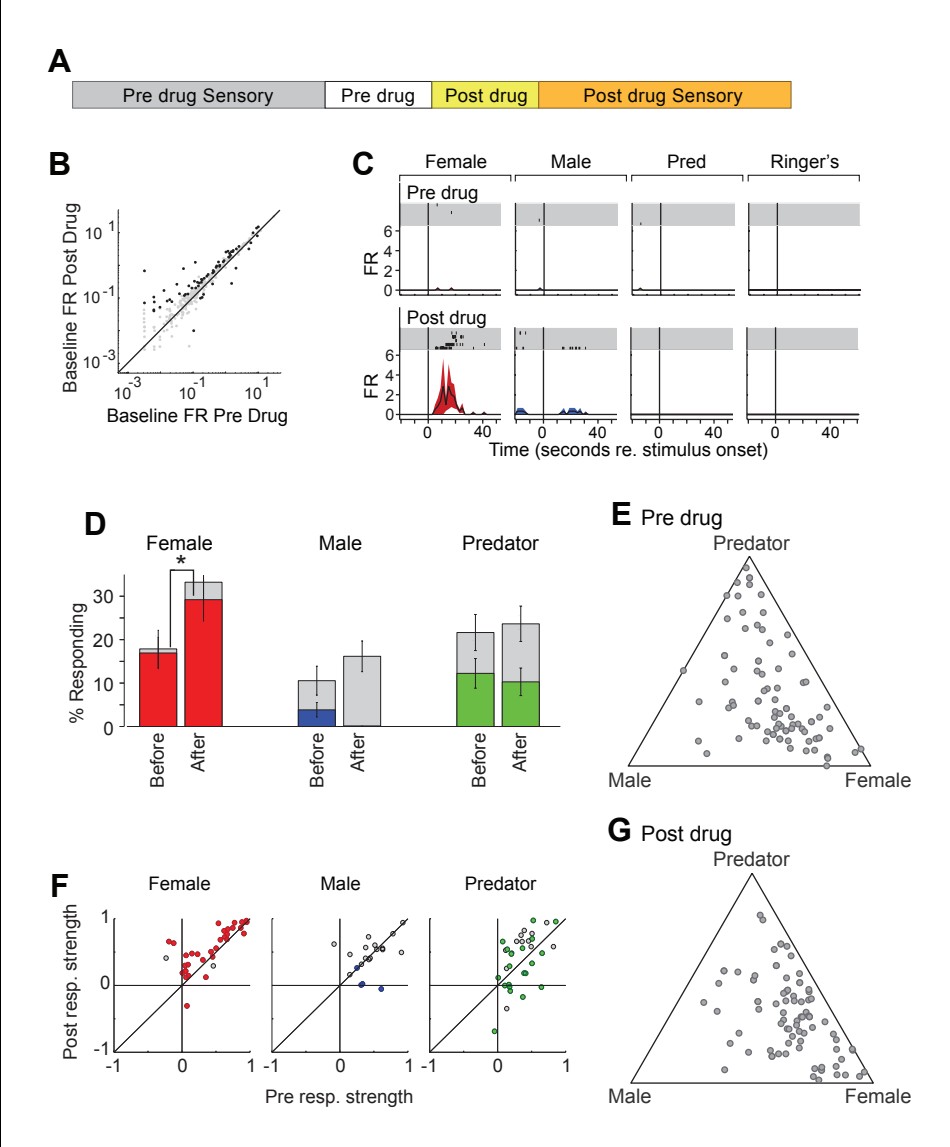

**Figure 7.** Acute chemogenetic activation of OXT signaling alters social interaction preference and neuronal response profile in the MeA. (**A**) Structure of combined pharmacology and electrophysiology experiments. Sensory responses were determined prior to drug application (~50 min), followed by a 10 min baseline period. Drugs were injected IP, and a second baseline period was collected. Finally, sensory responses were determined after drug application. (**B**) Firing rate (spikes/second) for single units in the absence of sensory stimuli (abscissa: firing rate before CNO; ordinate: firing rate after CNO; log scale). Black points indicate single units that displayed a significant change in activity after CNO injection. (**C**) Responses of a single MeA unit to sensory stimuli before and after the mouse was injected with CNO. For each panel, raster plots indicating the timing of individual action potentials elicited by multiple presentations of the same stimulus (shaded region). Histograms of the mean response and standard error for the same data are shown below each raster plot. Time zero indicates alignment to the start of stimulus presentation. (**D**) Comparison of responses to male, female, and predator stimuli before and after CNO injection. Grey bars: percentage of single MeA units that responded to each stimulus (nonparametric ANOVA; $p<=0.05$). Colored bars: percentage of single units that responded most strongly for the indicated stimulus. Asterisks indicate a significant difference in the fraction of responsive units before and after CNO injection (permutation t-test; *$p<0.05$). (**E, G**) Selectivity of MeA units to sensory stimuli before (**E**) and after CNO injection (**G**). Each point represents the sensory responses of an individual unit with at least one significant response to male, female, or predator stimuli. (**F**) Comparison of response strength before and after CNO injection for all single units responding to a given stimulus. Grey points indicate single units with a statistically significant response to the stimulus plotted in each axis. Colored points indicate units with a significant response

*Figure 7 continued on next page*

*Figure 7 continued*

that was also the strongest response (compared with responses to the other stimuli). Abscissa: (post-pre)/ (post + pre) prior to CNO injection. Ordinate: (post-pre)/(post + pre) after CNO injection.

DOI: https://doi.org/10.7554/eLife.31373.014

The following figure supplement is available for figure 7:

**Figure supplement 1.** Acute chemogenetic activation of OXT signaling alters social interaction preference.

DOI: https://doi.org/10.7554/eLife.31373.015

neural responses. This suggests that the impact of PVN-derived OXT on MeA sensory processing is largely restricted to opposite sex stimuli.

Interestingly, electrophysiological recordings enabled us to assess the effect of DREADD activation of OXT neurons in the PVN more immediately than was possible in our behavior tests. The enhancement of female stimulus evoked responses by chemogenetic activation was most pronounced immediately after CNO injection and declined during subsequent stimulus presentations. The time range during which DREADD activation most strongly influenced sensory responses was marked by intense grooming that prohibited tests of social preference in the corresponding behavior experiments. These findings suggest that DREADD activation initially impacted MeA responses strongly, but led to a subsequent inactivation of OXT signaling. In support of this hypothesis, a second dose of CNO administered at the end of each electrophysiology experiment did not noticeably influence the firing rate of MeA units (data not shown). These results are consistent with a relatively short duration (~15 min) of CNO driven enhanced OXT activity, followed by a long-lasting inhibition of OXT signaling, which may result from depletion of presynaptic OXT and/or desensitization of MeA OXTR.

## Discussion

In mice, social preference for male and female conspecifics has been shown to emerge during puberty, resulting in male mice preferentially investigating female stimuli over male stimuli (*Drickamer and Brown, 1998*). Here, we have explored the neuronal basis for this behavior and uncovered, at both behavioral and neuronal activity levels, a key role of the neuromodulator OXT in the discrimination of sex-specific cues by males. We show that loss of OXT in adult males leads to loss of social preference for female over male mice. Our experiments demonstrate that OXT signaling via expression of OXTR, but not AVP receptors, in a molecularly defined neuronal population of neurons in the MeA is necessary and sufficient for the behavioral and neuronal discrimination of sex-specific cues. We also show that OXT signaling acutely regulates adult male social preference by rapidly modulating the sensory responses of aromatase-expressing neurons to social cues, rather than by shaping circuit components during development. Reversible inhibition or activation of OXT-expressing neurons in the PVN, a main source of intracerebral OXT, leads to corresponding changes in the neuronal activation patterns in response to male and female chemostimuli, suggesting that OXT enables the dynamic modulation of neural circuits required for the discrimination of sex-specific cues, thus tailoring the display of social preference for male and female conspecifics to the current context.

### Distinct and overlapping functions of OXT and AVP in social discrimination

AVP and OXT regulate overlapping social behaviors with the specific function of the two peptides differing according to the sex and age of the animal, and the target neuronal population (*Johnson and Young, 2017*; *Stoop, 2012*). For example, early studies using the monogamous prairie voles suggest that OXT and AVP act to facilitate the formation of pair-bonding in both male and female voles, but male voles are more sensitive to AVP manipulations while female voles are more sensitive to OXT manipulations (*Cho et al., 1999*; *Young et al., 2011*). OXT and AVP have also been shown to act via different neural circuits to regulate a given behavior. For instance, mice with impaired OXT or OXTR function and mice with genetic ablation of V1aR all present impairment in social recognition of familiar and novel conspecifics (*Bielsky et al., 2004*; *Ferguson et al., 2001*; *Ferguson et al., 2000*; *Wersinger et al., 2008*). However, pharmacological manipulation using

receptor antagonists and virus-mediated rescue demonstrate that signaling critical for social recognition acts via OXTR in the MeA for, and via V1aR in the lateral septum (*Bielsky et al., 2005*; *Ferguson et al., 2001*).

Our results reveal a robust specificity in the role played by OXT in regulating the sensory processing of sex-specific social cues in males. We extend previous findings by showing that in the 3-chamber-social-investigation paradigm OXT, but not AVP, plays a unique role in the discrimination of sex-specific cues by males and that this signaling is exclusively mediated by OXTR in the MeA, and does not involve V1aR nor V1bR. Moreover, despite the well-established cross reactivity of OXT and AVP on all of OXTR, V1aR and V1bR, and the presence of rich AVP fiber tracks in the MeA, it appears that endogenous AVP does not compensate the defects in social interactions resulting from loss of OXT.

## The role of OXT in MeA processing of sex-specific cues in female mice

Our behavioral experiments revealed little bias in the time female mice spent interacting with male versus female mice. We note that this result differs from previous reports in which a preference of intact female mice for male-derived chemo-stimuli or intact males was demonstrated (*Bakker et al., 2002*; *Brock and Bakker, 2011*; *Chalfin et al., 2014*; *D'Udine and Partridge, 1981*; *Ramm et al., 2008*). This discrepancy may result from differences in the genetic backgrounds of the mouse lines or slight differences (e.g., stimulus source) in the behavioral paradigms used by each study. Regardless, we found that presence or absence of OXT did not noticeably impact the preference of female mice for male versus female stimuli. This suggests that OXT plays a sex-specific role in sculpting the social preferences of male versus female mice. OXT has been shown to lead to sexually dimorphic behavioral outcomes due to the sex-specific sensitivities of local circuits to hormones (*Li et al., 2016*; *Nakajima et al., 2014*). In our study, the sex specificity in OXT function coincides with a sexual dimorphism in MeA organization and function. The site of OXT action has been mapped to the MeApd aromatase-expressing neurons, with more MeApd aromatase-expressing neurons in males than females (*Wu et al., 2009*). The observed sex-specific role of OXT in social preference might be due to gender-specific differences in MeA physiology or circuitry. In further support of this idea, OXT was also found to selectively affect the plasticity of neuronal population activity in the MeA of awake behaving males, but not females, after sexual experience (*Li et al., 2016*). Since we observed no clear effect of OXT on social preference displayed by females, we focused our subsequent studies on male mice where the impact of OXT on social preference was clear and reliable.

## The role of OXT in MeA processing of sex-specific cues in male mice

OXT has been implicated in the modulation of a large number of social behaviors, including social recognition and social learning (*Ferguson et al., 2001*; *Ferguson et al., 2000*; *Takayanagi et al., 2005*; *Wersinger et al., 2008*), the formation of pair bonds (*Young et al., 2011*), parental care (*Takayanagi et al., 2005*), social defeat (*Guzmán et al., 2013*) and reinforcement of social reward (*Dölen et al., 2013*). Recently, impaired OXT function has been implicated in the abnormal social communications seen in certain human psychiatric diseases such as autism and schizophrenia, although a causal role of OXT in specific human behaviors and associated disorders is still to be clearly established (*De Berardis et al., 2013*; *Gordon et al., 2013*; *Heinrichs and Domes, 2008*; *Owen et al., 2013*; *Pobbe et al., 2012*). In rodents, chemosensory cues emitted by individuals are essential to encode the animal's social and physiological status, in turn leading to specific social behavioral responses between conspecifics such as mating, fighting or parenting. OXT mutant mice have been shown to display impaired social memory and altered responses to socially relevant chemosensory cues in the MeA and in MeA projections (*Ferguson et al., 2001*), supporting the notion that OXT modulates social behaviors by directly affecting the MeA processing of chemosensory cues encoding social information. Importantly, previous studies and the experiments we report here suggest that sensory modulation by OXT is specific for social cues and does not affect the detection of odor cues not relevant for social behavior.

To further investigate mechanisms underlying the role of OXT in male mouse social preference, we used genetic and viral tools to conditionally impair OXT signaling in distinct brain regions and neuronal populations, and mapped the site of OXT action to the sexually dimorphic aromatase-expressing neurons of the MeA. Furthermore, virus-mediated expression of OXTR in the MeA

rescues sex-specific odor discrimination impairment in *Cyp19a1-Cre;Oxtr*<sup>flox/flox</sup> male mice, whereas expression of OXTR in the BNST alone has no effect, thus helping to further delineate the site of OXT action to aromatase-expressing neurons of the MeA.

This in turn led us to perform in vivo multisite extracellular recordings and directly monitor the responses of individual MeA neurons to conspecific and predator stimuli with and without OXT signaling. Our results showed that, in the absence of OXTR, MeA neurons display overall reduced responses to chemosensory cues, while the percentage of neurons responding to male cues is increased, suggesting that the lack of preference of OXT mutant mice for sex-specific cues results from abnormal patterns of MeA neural activity. Moreover, consistent with a specific role of OXT in the processing of conspecific cues, neuronal responses to mouse cues are reduced more than those to predator cues in OXT mutants. Importantly, the observed behavioral and neural phenotype observed in OXT mutants may result from defects in early OXT-dependent developmental processes, or from the acute lack of OXT-mediated signaling. Our data showing that similar sex-preference behavioral defects are observed in constitutive as well as in conditionally virus-mediated OXTR ablation in the adult MeA strongly suggest a role for OXT signaling in adult. This conclusion is further supported by the ability of OXTR-expressing viral injection in the adult MeA, but not BNST, to rescue the phenotype of OXTR mutants, and by the observation of behavioral and neural phenotypes resulting from acute reduction of OXT activity in adults using chemogenetic or pharmacological approaches. Therefore, the effect of OXT mediated through OXTR in aromatase-expressing neurons is not developmental, but rather acts on a moment-to-moment basis even in adults.

Our previous study showed that the establishment of the sexually dimorphic sensory representation in the MeA requires steroid signaling near the time of puberty to organize the functional representation of sensory stimuli (*Bergan et al., 2014*). In contrast to the organizational effect of steroids, we found here that the OXT-mediated acute modulation of MeA neuronal response can alter social behavior in the adult on a moment-to-moment basis, while maintaining the integrity of circuit connectivity. Social recognition is similarly modulated by acute variation in OXT signaling (*Ferguson et al., 2001*). Our results further highlight the important role of neuropeptides in allowing the functional flexibility of neuronal circuits driving genetically pre-programmed instinctive behaviors.

Recent studies have shown that specific activation of aromatase-expressing neurons in the MeA promotes aggressive behaviors (*Unger et al., 2015*), and that reproductive sensory cues are processed by both the MeApv (*Ishii et al., 2017*) and the MeApd (*Bergan et al., 2014*; *Choi et al., 2005*). The overlapping control of aggressive and reproductive behaviors in the MeA mirrors the neuroethological argument that aggression is best understood as a component of reproductive behavior (*Tinbergen, 1951*). In further support of this view, our data show that OXT signaling through MeA aromatase-expressing neurons modulates the preference of male mice for interacting with female versus male mice, thus mediating a change in social preference that is likely to alter the balance of reproductive versus aggressive behaviors.

The change in sensory responses observed in the MeA parallels the changes observed in male social preference, suggesting that the observed behavioral effect of OXT signaling is achieved by filtering which sensory responses have access to behavioral centers in the hypothalamus on a moment-to-moment basis. We showed that OXT enhances the responses of neurons in the MeA that encode female stimuli while suppressing sensory responses to both male and predator stimuli. This increased salience of female stimuli observed in this study is reminiscent of the strengthened responses for attended auditory and visual stimuli (*Desimone and Duncan, 1995*; *Winkowski and Knudsen, 2006*). OXT is dynamically regulated by the external environment and an animal's internal physiological state (*Devarajan and Rusak, 2004*; *Higashida et al., 2017*; *Kalin et al., 1985*; *Kendrick et al., 1986*; *Wathes et al., 1992*), and the OXT-mediated filtering of sensory cues shown in our study likely supports a fine-tuning of behavioral responses according to the social context.

## Mechanism of OXT signaling in the MeA

An intriguing question raised by our data is how is the specificity of OXT action to conspecific responses being achieved in the MeA. Our study identifies aromatase-expressing neurons in the MeA as a necessary and sufficient site for OXT to shape the social preference of male mice to explore female stimuli. OXT has been shown to regulate specific brain functions by facilitating the release of other neurotransmitters, in particular dopamine and serotonin (*Dluzen et al., 2000*;

*Dluzen et al., 1998*; *Dölen et al., 2013*; *Liu and Wang, 2003*; *Olazábal and Young, 2006*; *Ross et al., 2009*). In addition OXT has been documented to increase neuronal firing rates, mainly, though not exclusively in fast-spiking interneurons, resulting in lower background activity and enhanced information transfer in cortical and non-cortical circuits (*Marlin et al., 2015*; *Oettl et al., 2016*; *Owen et al., 2013*; *Xiao et al., 2017*). These mechanisms may similarly operate in the MeA, and understanding how OXT signaling in aromatase-expressing neurons alters social preferences and the representation of social cues is an important line of research for future studies.

Altogether, our results uncover aromatase-expressing MeA neurons as a hub for OXT-dependent modulation of female-evoked neural responses and behaviors in the male mouse. OXT selectively increases MeA responses to female stimuli, indicating that OXT signaling has the ability to differentially modulate the specific sensing of female cues through its action on MeA neurons co-expressing aromatase and OXTR. The MeA receives direct sensory input from the AOB and is reciprocally connected with hypothalamic behavior centers. Although sex-specific sensory cues are relayed separately at the receptor level, AOB mitral cells, which send input to MeA neurons, often respond to multiple sensory stimuli. It is possible that MeA neurons co-expressing aromatase and OXTR receive input from AOB neurons that are preferentially responsive to female stimuli. Alternatively, the specificity of OXT in modulating female-evoked neural responses may be achieved by regulating the local MeA circuitry, for example by acting on a specific set of local aromatase- and OXTR-expressing interneurons that boost responses to female cues while dampening responses to male and predator signals. These possibilities are not mutually exclusive. Understanding the molecular identity and circuit connectivity of neurons both directly and indirectly modulated by OXT will be essential for understanding how OXT regulates female-evoked neural responses and behaviors in male mice.

The data presented here provide significant new insights into the role of OXT signaling in the control of social behavior. OXT selectively enhanced neural and behavioral responses to female stimuli, suggesting that subsets of female-responsive and male-responsive neurons in the MeA likely express different levels of OXTR. Thus, by dynamically and selectively tuning the neural representation of social cues in a genetically defined neuronal population of the MeA, OXT may alter the behavioral significance of social stimuli on a moment-to-moment basis.

## Materials and methods

### Animals

All mice were maintained in 12 hr:12 hr light:dark cycles with food and water available *ad libitum*. Animal care and experiments were carried out in accordance with the NIH guidelines and approved by the Harvard University Institutional Animal Care and Use Committee.

The conditional oxytocin receptor knockout mice (B6.129 (SJL)-*Oxtr*$^{tm1.1Wsy}$/J, stock number: 008471) were from Jackson laboratories. The line had been backcrossed to the C57BL/6J inbred mice for at least 11 generations. V1aR knockout mice (B6.129P2-*Avpr1a*$^{tm1Dgen}$/J, stock number: 005776) and V1bR knockout mice (B6;129 $\times$ 1-*Avpr1b*$^{tm1Wsy}$/J, stock number: 006160) were obtained from Jackson laboratories and were backcrossed to the C57BL/6J inbred mice for at least six generations in our facility. The *Oxt-iCre* and *Trh-Cre* transgenic mouse lines were previously reported (*Krashes et al., 2014*). The lines were previously maintained on a mixed C57BL/6 $\times$ 129S background and were backcrossed to C56BL/6J mice for our study.

The *Cyp19a1-Cre* line was generated by BAC recombination, such that the Cre coding sequence was inserted in front of the start codon of the *Cyp19a1* gene, which encodes aromatase. The ATG codon of the *Cyp19a1* gene in the BAC was additionally mutated into a TTG stop codon.

The absence of germline recombination events due to the presence of Cre and floxed alleles together in the germline was confirmed by two genotyping strategies. One pair of genotype primers (P1: 5′-TGAAGAAGGATGGGCTTTTG-3′; P2: 5′- GGTCCCAGGAAAGAGTCAGC-3′) was designed to amplify both WT and floxed loci of the conditional OXTR knockout allele (244 bp and 341 bp, respectively). Another pair of primers (P1; P3: 5′- TGGGAGTCCAGAGATAGTGGAA-3′) was designed to amplify the recombined locus (485 bp). The germline Cre-mediated deletion of floxed loci can be detected due to the lack of PCR products corresponding to the floxed loci (341 bp), and the presence of PCR products corresponding to the recombined locus (485 bp). We did not observe any germline recombination in our crosses.

The absence of germline recombination and partial recombination due to activity in the early embryo was additionally confirmed for the *Cyp19a1-Cre* line by generating a double transgenic line, consisting of the *Cyp19a1-Cre* line and the Rosa26-lsl-tdTomato reporter. Mice containing both *Cyp19a1-Cre* and Rosa26-lsl-tdTomato were bred to maintain the double-transgenic line. The expression of tdTomato was observed for multiple generations: no germline-mediated recombination was observed in this double-transgenic line, and the tdTomato expression pattern faithfully reflected the endogenous expression pattern of the aromatase gene, thus excluding the possibility of germline recombination or partial recombination.

## Stereotaxic injection of AAV virus

The AAV-GFP and AAV-GFP-Cre viruses of serotype one were ordered from Penn Vector Core; the AAV-hSyn-DIO-hM4Di-mCherry virus and AAV-hSyn-DIO-hM3Dq-mCherry of serotype eight were obtained from the University of North Carolina Vector Core. To generate AAV-OXTR-IRES-GFP virus, the CMV-OXTR-IRES-GFP-BGH PolyA cassette was inserted into the XbaI site of pSub801, and the resulting plasmid was used to generate high titer virus of serotype one in the Harvard virus Core.

Mice 6–8 weeks old were anesthetized by ketamine/xylazine. A total of 200 nl virus was stereotaxically injected into the target areas. The coordinates used for the PVN injections were: bregma: anterior-posterior, –0.90 mm; dorsal-ventral, –4.75 mm; lateral,±0.25 mm, those for the SON were: bregma: anterior-posterior, –0.80 mm; dorsal-ventral, –5.50 mm; lateral,±1.3 mm, those for the posterior medial amygdala were: bregma: anterior-posterior, –1.90 mm; dorsal-ventral, –5.0 mm; lateral,±1.75 mm, and those for the BNST were: bregma: anterior-posterior, –0.34 mm; dorsal-ventral, –4.50 mm and −4.00 mm; lateral, ±0.75 mm. Mice recovered for 4 weeks after surgery before being individually caged for social behavior tests.

## Immunohistochemistry

Mice brains were dissected and fixed overnight in 4% paraformaldehyde at 4°C, and 50 μm sections were prepared using a vibratome. Brain sections were blocked in PBX (PBS with 0.05% Triton X-100) containing 10% FBS for 1 hr at room temperature, following by overnight incubation in the appropriate primary antibodies at 4°C. The sections were then washed with PBX, and incubated with either Fluorophore-conjugated secondary antibodies or Biotin-SP-labeled secondary antibodies for 1 hr at room temperature. The sections were then incubated with fluorophore-labeled streptavidin when necessary, and mounted after washing in PBX.

The following primary antibodies were used: rabbit anti-oxytocin (Immunostar, AB911, 1:1000), rabbit anti-vasopressin (Immunostar, 20069, 1:1000), and rabbit anti-GFP (Invitrogen, Waltham, MA, A-11122; 1:1000). Biotin-SP-AffiniPure Goat Anti-Rabbit IgG (111-065-144), and CY3 streptavidin (016-160-084) were from Jackson ImmunoResearch Laboratories, West Grove, PA. Alexa Fluo 488 Anti-mouse IgG, Alexa Fluo 488 streptavidin was from Invitrogen, Waltham, MA, (Catalog Number: S11223).

## In-situ hybridization

RNA in situ hybridization was performed as described previously (*Isogai et al., 2011*). Briefly, mice brains were dissected and immediately embedded in OCT, and 20 μm sections were prepared using a cryostat. Brain sections on slides were washed with PBS once, fixed in 4% paraformaldehyde for 10 min at room temperature, washed three times with PBS, treated with acetylation solution (0.1 M triethanolamine with 2.5 μl ml$^{-1}$ acetic anhydride) for 10 min, and then incubated with the pre-hybridization solution (50% formamide, 5 × SSC, 5 × Denhardt's, 2.5 mg ml$^{-1}$ yeast RNA, 0.5 mg ml$^{-1}$ herring sperm DNA) for 2 hr. The slides were then added with the hybridization buffer containing the appropriate DIG or FITC labeled probes and subsequent hybridization was carried out in a sealed chamber at 68°C overnight. The brain sections were then washed sequentially in 5XSSC, 0.2XSSC and TNT buffer (100 mM Tris, pH 7.5, 150 mM NaCl, 0.05% Tween 20). After the posthybridization washes, slides were incubated with anti-FITC-POD (Roche, 1:250) and/or anti-DIG-POD (Roche, 1:500), followed by TSA amplification (Perkin Elmer; Waltham, MA).

## CNO injection and behavioral analysis

A stock solution was made by dissolving 5 mg of CNO (Sigma Aldrich, St. Louis, MO) in 1.333 ml dH2O to make 3.75 mg/ml solution (10 mM). A working solution was prepared by diluting the working solution with normal saline at a ratio of 1:100. Mice received IP injection of CNO at a dose of 0.3 mg/kg, followed by behavioral analysis. To observe acute behavioral effects, mice were injected with CNO once, followed by behavioral test one hour later.

## OXTR antagonist injection

The OXTR antagonist (OXTR-A, L-368,899 hydrochloride) was purchased from Tocris Bioscience, Bristol, United Kingdom. A 10 mg/ml stock solution of OXTR-A was prepared in dH2O, and was delivered by intraperitoneal injection at a dosage of 5 mg/kg$^{-1}$, followed by behavioral analysis and electrophysiological recording.

## Behavioral analysis

All behavioral tests were performed in the dark phase of the light/dark cycle. Mice aged between 2 to 4 months were used for behavioral tests.

### Odor preference

Female or male beddings were collected from group housed C57BL6 mice on the same day of the odor preference test and the beddings of the same sex cages were not changed for 72 hr. Subject mice were individually caged for one week before behavioral test. About 50 ml of beddings were taken using 50 ml conical tubes and used for each odor preference assay. Female and male beddings were presented simultaneously to the resident mouse in its home cage. The two piles of bedding were separated by a plastic divider. The subject mouse was allowed to investigate the bedding for 5 min and the entire duration of the test was videotaped. The time spent by the test mouse to investigate each pile of beddings was quantified using the Observer software (Noldus,Leesburg, VA). The odor preference score was calculated as (Time spent in investigating female bedding-Time spent in investigating male bedding)/(Time spent in investigating female bedding +Time spent in investigating male bedding).

### Odor habituation/dishabituation test

Urine was collected from five adult group-housed C57BL/6 males and five adult group-housed C57BL/6 female. Urine samples from the same sex were pooled and stored in at −80°C. Before testing each day, aliquots of urine were thawed and used. Subject mice were housed alone for a minimum of 3 days before the odor habituation/dishabituation test. The subject mice first received three 2 min presentations of saline at 1 min intervals, followed by three 2 min presentations of one urinary odor and finally by three 2 min presentations of a second urinary odor. Fisher's LSD test was used to compare the difference between the durations that resident spent investigating the stimuli.

### 3-Chamber social preference assay

A 111.8 cmx111.8 cm plexiglass arena was divided into 3 chambers of equal size. The left and right chambers each contained a flipped pencil holder (enclosure cage) with a 200 ml glass beaker on top. To test social preference, the two doors leading to the side chambers remained closed initially and the male mouse was allowed to acclimate to the central chamber for 10 min. In the habituation stage of the assay, the pencil holders in the side chambers were empty and the mouse was allowed to explore the side chambers for 10 min. In the social interaction phase, an unfamiliar C57BL6 mouse was placed in one of the cages and a second C57BL6 mouse of opposite sex was placed in the other cage. The subject mouse was then allowed to explore the arena for 10 min. The motion of the subject mouse during the habituation and social interaction phases of the assay was automatically tracked by Ethovision (Noldus). The time spent by the mouse in each chamber and in each interaction zone and the total distance travelled were calculated automatically by Ethovision. The interaction zone was defined as the area within 5 cm from the border of the enclosure cage. The social interaction preference score was calculated as (Time spent in female interaction zone-Time spent in male interaction zone)/(Time spent in female interaction zone +Time spent in male interaction zone).

To observe the acute effect of OXTR-A on social preference of male mice, subject mice were first tested in the 3-chamber social preference paradigm after saline injection, and then the preference was tested again 30 min after mice received IP injection of OXTR-A.

## Y-Maze odor discrimination assay

The subject mouse was allowed to acclimate to the starter chamber for 10 min. In the habituation stage of the assay, the left and right goal zones contained only an empty 10 cm petri dish and the mouse was allowed to explore Y-maze for 10 min. In the test phase, odor stimulus pairs were added to the petri dishes in either the left or right goal zones. For experiments involving OT mutants and *Cyp19a1-Cre; Oxtr^flox/flox^*, a clean food pellet or one spiked with 25 µl of fox urine (PredatorPee, Lincoln, Maine) served as the odor stimuli pair and were placed on the petri dishes in the left and right goal zones. For experiments involving *Trh-Cre; Oxtr^flox/flox^*, 20 µl of female mouse urine or fox urine was added to a cotton pad and then placed on the petri dish. The subject mouse was allowed to explore the Y-maze for 10 min. The motion of the subject mouse during the habituation and test phases of the assay was tracked and the duration spent by the mouse in each goal zone was calculated automatically by Ethovision.

## Statistical analysis of behavioral data

All behavioral comparisons were analyzed using the Prism6 software to determine the statistical significance of behavioral results. Statistical tests for odor and social preference were performed by comparing the bias in time an animal spent near stimulus A (ex. female conspecific) versus stimulus B (ex. male conspecific) to the bias in time spent in the same physical area without the respective stimuli (ex. empty cups). In all cases, a family-wise error rate correction for multiple comparisons was made using the Holm-Sidak's method. Thus, our statistical tests account for any observed bias in an animal's preference for one stimulus while accounting for any innate spatial biases the animal may display.

## VNO stimulation and electrophysiology

Mice were anesthetized, tracheotomized, and prepared for electrophysiological recording as previously described (*Ben-Shaul et al., 2010*; *Bergan et al., 2014*). A stimulating cuff electrode was placed around the rostral sympathetic nerve trunk to control the VNO pump and access of stimuli to the VNO sensory epithelium. A ~ one mm² craniotomy was opened dorsal to the MeA based on stereotaxic coordinates, to allow the insertion of Neuronexus probes (NeuroNexus Technologies, Ann Arbor, Michigan: a1 × 32–10 mm 50-500-413) coated with a fluorescent dye (DiI or DiD; Invitrogen, Carlsbad, CA) into the MeA (*Bergan et al., 2014*). Accurate stereotaxic targeting to the MeA was confirmed by post mortem histological analysis of the electrode tract.

Urine was collected from adult singly housed estrus female and singly housed male mice of the balb/C, CBA, or C57Bl6 strains and immediately placed in liquid nitrogen, for subsequent storage in −80C. Urine samples from fox, mountain lion, and bobcat were obtained from PredatorPee (Lincoln, Maine). All sensory stimuli were presented at least six times in a randomized order for each dataset. VNO stimuli were applied by placing 1 ul of stimulus (1:100 dilution in Ringer's) directly into the nostril, followed by activation of the sympathetic nerve cuff electrode to facilitate VNO pumping and stimulus entry to the VNO lumen. The nasal cavity and VNO was cleaned with Ringer's solution after each stimulus presentation (*Ben-Shaul et al., 2010*).

Animals were maintained under anesthesia while CNO or the OXTR antagonist were injected (see above for additional injection details), and the electrophysiological recording was maintained throughout to allow identification of electrode movement due to the injection. Only units that were maintained throughout the recording were analyzed.

All recording channels were band-pass filtered (300–5000 Hz) and digitized at 25 kHz using an RZ2 processor, PZ2 preamplifier, and two RA16CH head-stage amplifiers (TDT, Alachua, FL). Custom MATLAB (Mathworks, Natick, MA) scripts (https://github.com/joebergan/Spike-sorting) were used to extract 3.5 ms spike waveforms from the continuous data (*Bergan and Ben-Shaul, 2017*). A copy is archived at https://github.com/elifesciences-publications/Spike-sorting. Waveforms were extracted for the eight nearest electrode channels and units were typically visible on 2–4 contiguous electrode channels, which enhanced the isolation of single units. Single unit clusters were determined based

on the principal components using KlustaKwik (*Harris et al., 2000*; *Bergan et al., 2014*) and manually verified, and adjusted using Klusters (*Hazan et al., 2006*).

Single unit spike clusters were determined based on spike shapes, projections on principal component space (calculated independently for each recording session), and autocorrelation functions (interspike interval). Single units displayed a distinct spike shape that was fully separated from both the origin (noise) and other clusters (multi-unit) with respect to at least one principal component projection, and also displayed a clear refractory period in the interspike interval histogram.

Valid sensory responses were identified by comparing the spike rate after sensory stimulation to the spike rate immediately preceding sensory stimulation for each unit. Responsive units passed a non-parametric ANOVA statistical test performed at the significance level of $p \leq 0.01$. Response magnitude was quantified as the change in average firing rate during the 40 s following stimulus presentation relative to the firing rate during the 20 s prior to stimulus presentation. Unless otherwise noted, statistics comparing two populations of units were performed using a nonparametric permutation test (*Efron and Tibshirani, 1993*) as observed distributions were not typically normally distributed.

## Additional information

### Competing interests
Catherine Dulac: Senior editor, *eLife*. The other authors declare that no competing interests exist.

### Funding

| Funder | Grant reference number | Author |
| --- | --- | --- |
| Howard Hughes Medical Institute | | Catherine Dulac |
| National Institute on Deafness and Other Communication Disorders | R01DC013087 | Catherine Dulac |
| Simons Foundation | Simons Foundation Autism Research Initiative - 308094 | Catherine Dulac |

The funders had no role in study design, data collection and interpretation, or the decision to submit the work for publication.

### Author contributions
Shenqin Yao, Joseph Bergan, Conceptualization, Data curation, Formal analysis, Validation, Investigation, Visualization, Methodology, Writing—original draft; Anne Lanjuin, Formal analysis, Methodology; Catherine Dulac, Conceptualization, Resources, Supervision, Funding acquisition, Project administration, Writing—review and editing

### Author ORCIDs
Shenqin Yao http://orcid.org/0000-0003-2992-4752
Joseph Bergan http://orcid.org/0000-0002-9386-2595
Catherine Dulac http://orcid.org/0000-0001-5024-5418

### Ethics
Animal experimentation: Animal care and experiments were carried out in accordance with the NIH guidelines and approved by the Harvard University Institutional Animal Care and Use Committee (protocol numbers: 23-12, 25-13, 97-03)

### Decision letter and Author response
Decision letter https://doi.org/10.7554/eLife.31373.018
Author response https://doi.org/10.7554/eLife.31373.019

## Additional files

**Supplementary files**
• Transparent reporting form
DOI: https://doi.org/10.7554/eLife.31373.016

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
