## [Decision Letter]

Thank you for submitting your article "Oxytocin Signaling in the Medial Amygdala is required for Sex Discrimination of Social Cues" for consideration by *eLife*. Your article has been favorably evaluated by Huda Zoghbi (Senior Editor) and three reviewers, one of whom, Richard D Palmiter (Reviewer #1), is a member of our Board of Reviewing Editors. The following individual involved in review of your submission has agreed to reveal their identity: W. Scott Young (Reviewer #3).

The reviewers have discussed the reviews with one another and the Reviewing Editor has drafted this decision to help you prepare a revised submission.

Summary:

This paper provides important advances in understanding how oxytocin affects social behavior by identifying post-synaptic, aromatase/oxytocin receptor neurons in the medial amygdala that regulate perception of sex-related odors. Before final acceptance, a few concerns need to be resolved.

Essential revisions:

1) Behavioral alterations are mapped to circuit alterations but without clear causal explanations for how these behavioral effects are mediated. The finding that OXT specifically alters the responses to female stimuli are interesting, but the claim that the effects of acute OXTR blockade are surprising is not well supported. The electrophysiological analysis of the responses of MeA neurons provides some insight but questions about why the alteration of responses is sex-specific receive little attention and I believe that these are among the most interesting findings of the manuscript.

2) Authors should emphasize the controls that were used when using DREADD and CNO to rule out non-specific effects of CNO.

3) The possibility that genetic background might influence the role of vasopressin receptors should be addressed.

4) The authors should do social preference tests after the intense grooming subsides when using the excitatory DREADD as suggested.

5) The authors should discuss their results in broader context, if possible. What is the overall impact of poor social recognition by males? Does this lack of preferential recognition of females affect mating behavior, fertility or post-partum behaviors of males?

6) Authors should indicate what experiments they did to be sure that all of the mice used for experiments did not have unexpected recombination events due to having Cre and floxed alleles together in the germline.

---

## [Author Response]

Essential revisions:1) Behavioral alterations are mapped to circuit alterations but without clear causal explanations for how these behavioral effects are mediated. The finding that OXT specifically alters the responses to female stimuli are interesting, but the claim that the effects of acute OXTR blockade are surprising is not well supported. The electrophysiological analysis of the responses of MeA neurons provides some insight but questions about why the alteration of responses is sex-specific receive little attention and I believe that these are among the most interesting findings of the manuscript.

We agree with these points and have modified the text accordingly:

- In absence of slice recording, which is well beyond the scope of the present study, any mechanistic scenario remains hypothetical, but we agree that a set of plausible mechanistic hypotheses for how the observed behavioral effects are mediated is needed. We have added specific text in the Discussion that details possible circuit-level scenarios (see also later point in this response).

- The dynamic regulation of OXT by external environment and internal physiological state is well-recognized (Devarajan and Rusak, 2004; Higashida et al., 2017; Kalin et al., 1985; Kendrick et al., 1986; Wathes et al., 1992), and the ability of acute OXTR blockade to modulate behaviors has also been reported (Ferguson et al., 2001). In our initial submission, our intention was to emphasize the acute effects of OXT in social preference in contrast to a potential role in development, and we did not mean to claim any novelty with respect to the acute effects of OT. We have now clarified the text accordingly.

Indeed, our past study (Bergan et al., *eLife* 2014) has demonstrated that the establishment of the striking sexually dimorphic sensory representation in the MeA requires steroid signaling near the time of puberty to organize the functional representation of sensory stimuli. In the present work, we show that virally expressed OXTR restores social preference in the adult, indicating a role of OXT in dynamically regulating social preference in adults, in contrast to an organizational role. Following the reviewer’s suggestion, we have clarified in the text that acute behavioral effects of OXTR blockage have been reported previously, and have phrased the conclusions from our current work accordingly.

- There are two aspects of sex-specificity in our study: (1) a sex-specificity in the sex of the tested animals, and (2) a sex-specificity in the response of the tested animals to cues from males versus females:

1) We have focused on the function of OXT in male mice because they reliably demonstrate a strong social preference to explore female mice. By contrast, female mice of all tested strains reliably displayed a small to non-significant social preference for females in similar behavioral assays. This is made more explicit in the Results and Discussion sections.

2) In male mice, the alteration of neuronal responses due to loss of OXT signaling differentially affects the responses to female versus male and predator cues. There are several potential mechanisms consistent with the observed results, which all involve the presence of distinct channels of sensory information for female stimuli, whose modulation by oxytocin occurs in aromatase-positive neurons and are separable from male- and predator-specific sensory sensing. We now describe candidate mechanisms more thoroughly in the Discussion.

We thank the reviewers for these suggestions, which have helped further clarify the significance and broader context of our results.

2) Authors should emphasize the controls that were used when using DREADD and CNO to rule out non-specific effects of CNO.

In view of recent findings on the effects of CNO and CNO metabolites on brain function, it is indeed imperative to have appropriate controls for any DREADD experiment. In our study, we compared groups in which DREADDs were expressed in different brain areas. CNO only induced changes in social preference when paired with DREADD expression in the PVN, but not in the SON – confirming that the effect of CNO is specific to DREADD infected OXT neurons in the PVN. We now further clarify the concerns of the specificity and function of DREADDs in the text as follows:

“Two sets of controls confirmed that the observed DREADD-mediated effects of CNO are specific to the activation or inhibition of OXT-expressing neurons. […] Second, the OXT-iCre male mice expressing the inhibitory DREADD in the SON presented normal preference towards female mice before and after CNO injection, excluding the possibility that the impaired social preference is due to an off-target effect of CNO or its metabolite clozapine (Gomez et al., 2017; Saloman et al., 2016).”

3) The possibility that genetic background might influence the role of vasopressin receptors should be addressed.

This is indeed an important point that we had failed to address in our initial submission. The OXT KO, Avpr1a and Avpr1b lines were from the Jackson Laboratory. All lines were backcrossed to C57BL/6 for >6 generations, in order to avoid the impact of different genetic background on social behaviors. We did not observe any impaired social preference of Avpr1a and Avpr1b knockout lines in our lines in the C57BL/6 background. This information has now been added to the text.

4) The authors should do social preference tests after the intense grooming subsides when using the excitatory DREADD as suggested.

We agree with the suggestion and have explored the behavioral effects of chemogenetic activation of OXT neurons after the initial grooming effects subside. The results are summarized below and were added to the manuscript.

The excitatory DREADD virus, AAV8-DIO-hM3Dq-mCherry, was stereotaxically injected into the PVN of OXT-iCre male mice, and times spent investigating a pair of lightly anesthetized male and female stimulus mice (Figure 7—figure supplement 1) were measured at various intervals after IP injection of saline or CNO (15min, 2 days, 1 week and 2 weeks after the initial injection).

After saline injection, OXT-iCre mice expressing excitatory DREADD in the PVN preferentially investigated the female stimulus mouse. By contrast, in experimental animals, central OXT levels were immediately enhanced after CNO injection, as indicated by lengthy hypergrooming observed post CNO injection. However, after the hypergrooming subsided (15 min after CNO injection) experimental animals showed a loss of female preference that did not recover even when tested 2 days, 1 week and 2 weeks after CNO injection (Figure 7—figure supplement 1).

These data contrast with the results obtained by extracellular recordings, in which we observed a significant increase in the response strength of MeA neurons to female stimuli after CNO injection (Figure 7). Interestingly, the fine time resolution of the electrophysiological recording enabled us to directly assess the effect of DREADD activation of OXT neurons in the PVN over time by detecting changes in neuronal responses minutes after chemogenetic activation. In turn, the observed timeline of CNO activity provided by the recording data gave us a better understanding of the results obtained in the behavioral assays.

In the electrophysiology experiments, a 5-minutes baseline recording was conducted post-injection, followed by 6 times of presentations of 4 stimuli, overall lasting around 25 to 30 minutes post-CNO injection (Figure 7). We found that the most significant enhancement in the responses of MeA neurons to female stimulus was observed in the stimulus presentations immediately following CNO injection. By contrast, the effect of DREADDs on responses diminished over the six trials: that is, the strongest effects on electrophysiological responses correspond roughly with the timing of intense grooming in behaving animals. We also performed a second CNO injection at the end of each recording experiment, and found that this second dose of CNO did not produce further change in the firing rate of MeA neurons (data not shown). These results are consistent with relatively short duration (<15 min) of CNO driven enhanced OXT activity, followed by a long lasting inhibition of OXT signaling, which may result from depletion of presynaptic OXT and/or saturation of MeA OXTR. We have added specific text in the results that clarifies the observed temporal dependence of chemogenetic activation, and the behavioral data.

5) The authors should discuss their results in broader context, if possible. What is the overall impact of poor social recognition by males? Does this lack of preferential recognition of females affect mating behavior, fertility or post-partum behaviors of males?

We have extended the discussion on this topic. Briefly, we believe that OXT signaling provides an increase in the saliency of certain social stimuli in certain social contexts. Thus, the enhancement of sensory responses of male mice toward females is similar to the enhancement of visual responses observed in visual attention or the enhanced auditory responses observed in auditory attention.

6) Authors should indicate what experiments they did to be sure that all of the mice used for experiments did not have unexpected recombination events due to having Cre and floxed alleles together in the germline.The authors should also attend to the minor, editorial comments.

The absence of germline recombination was confirmed by two genotyping strategies: One pair of genotype primers are designed to amplify both WT and floxed loci of the conditional OXTR knockout allele (244 bp and 341 bp, respectively). Another pair of primers are designed to amplify the recombined locus (485 bp). The germline Cre-mediated deletion of floxed loci can be detected due to the lack of PCR products corresponding to the floxed loci (341 bp), and the presence of PCR products corresponding to the recombined locus (485 bp). We did not observe any germline recombination in our crosses. This strategy can rule out germline recombination, but still cannot detect partial recombination due to early embryo activity. The above genotyping information has been added to the Materials and methods section.

The absence of germline recombination and partial recombination due to activity in early embryo is additionally confirmed for the aromatase-Cre line by generating a double transgenic line, consisting of the aromatase-Cre line and the Rosa26-lsl-tdTomato reporter. Mice containing both aromatase-Cre and Rosa26-lsl-tdTomato were bred to maintain the double-transgenic line. The expression of tdTomato was observed for multiple generations. We did not observe any germline-mediated recombination in this double-transgenic line, and the tdTomato expression pattern faithfully reflects the endogenous expression pattern of the aromatase gene, thus excluding the possibility of germline recombination and of partial recombination due to activity in the early embryo.